# Hazards in Products of Plant Origin Reported in the Rapid Alert System for Food and Feed (RASFF) from 1998 to 2020

**Marcin Pigłowski [1],*** and **Magdalena Niewczas-Dobrowolska [2]**

[1] Department of Quality Management, Faculty of Management and Quality Science, Gdynia Maritime University, Morska 81-87, 81-225 Gdynia, Poland
[2] Department of Quality Management, Institute of Quality Sciences and Product Management, College of Management and Quality Sciences, Cracow University of Economics, Rakowicka 27, 31-510 Kraków, Poland; niewczam@uek.krakow.pl
*   Correspondence: m.piglowski@wznj.umg.edu.pl

**Abstract:** The elimination or reduction of hazards in plants is an important part of the "From field to fork" strategy adopted in the European Green Deal, where a sustainable model is pursued in the food system. In the European Union (EU), the Rapid Alert System for Food and Feed (RASFF) is in place to provide information on risks in the food chain. The largest number of notifications in this system concerns plants, followed by products of animal origin and other products. The goal of the study was to examine RASFF notifications for products of plant origin with respect to hazard, year, product, notifying country, origin country, notification type, notification basis, distribution status and actions taken in 1998–2020. Data were extracted from the RASFF notifications' pre-2021 public information database. A cluster analysis using joining and the two-way joining method was applied. The notifications mainly concerned aflatoxins in pistachios from Iran, ochratoxin A in raisins from Turkey, pesticide residues in peppers from Turkey, okra, curry, rice from India, tea from China and India, and pathogenic micro-organisms in sesame from India, and also basil, mint and betel from Thailand, Vietnam and Lao Republic. To ensure the safety of food of plant origin, it is necessary to adhere to good agricultural and manufacturing practices, involve producers in the control of farmers, ensure proper transport conditions (especially from Asian countries), ensure that legislative bodies set and update hazard limits, and ensure their subsequent control by the authorities of EU countries. Due to the broad period and scope of the studies that have been carried out and the significance of the European Union in the food chain, the research results can improve global sustainability efforts.

**Keywords:** food safety; food hazards; plants; RASFF; cluster analysis

## 1. Introduction

A sustainable global future should consider food security and food safety, taking public health into account to achieve long-term sustainability. According to the World Health Organization (WHO) definition, food security exists "when all people, at all times, have physical and economic access to sufficient, safe and nutritious food to meet their dietary needs and food preferences for an active and healthy life". This is closely linked to economic growth, social progress, political stability and peace. It should be noted that food safety can be recognised as a component of food security, as this refers to the fact that food is safe to eat and does not pose a risk to human health [1]. Food safety should include the sustainable development of the agri-food sector [1,2]. Thus, both sustainability and future food security require the consideration of food safety [1,3].

The most important challenge to food security and food safety is the growing human population [4]. However, it is important to point out that this mainly concerns developing countries. Considering sustainability in the context of food, it is noteworthy that in developing countries, attention is focused on food security, and in developed countries, on food safety [5]. Given this discrepancy, it therefore seems important to pay attention to the

movement of food from developing countries to developed countries. In order to reduce the risk of foodborne disease hazards, developing countries that trade in food should have an integrated and inclusive development policy with regard to food security [6].

In the Sustainability Assessment of Food and Agriculture systems guidelines issued by the Food And Agriculture Organization of the United Nations (FAO), food safety is mentioned in the theme "Product quality & information" within the economic resilience dimension. In this document, food safety hazard is defined as "a biological, chemical or psychical agent in, or condition of, food with the potential to cause an adverse health effect" [7]. Among the biological agents, there are, for example, mycotoxins and pathogenic micro-organisms; chemical agents can include pesticide residues, and physical agents comprise foreign bodies [8].

According to the requirements for food safety included in the European law, food that is injurious to health is considered unsafe and should not be placed on the market [9]. Therefore, the Rapid Alert System for Food and Feed (RASFF) was established to provide information on risks in the food chain. During the period 1979–2020, the largest number of notifications in this system related to food of plant origin (more than 43%), followed by food of animal origin (30%), with the remaining notifications referring to other types of food, feed and food contact materials [8].

## 1.1. Characteristics of the RASFF

Currently, the legal basis for the operation of the RASFF is the Regulation (EC) No. 178/2002, laying down the general principles and requirements of food law, establishing the European Food Safety Authority, and setting up procedures in matters of food safety. This Regulation obliges each RASFF member to report to the European Commission with information on any serious health risks deriving from food or feed. The members of the system are the 27 countries of the European Union (EU), the European Commission, the European Food Safety Authority (EFSA), the European Free Trade Association Surveillance Authority (ESA), Norway, Liechtenstein, Iceland, and Switzerland [9,10].

Alert notifications are sent when food presenting a serious risk is already on the market, and also after the control at the external borders of the EU (in a broader sense, the European Economic Area (EEA)), if there is potential hazard, and when rapid action is required. The RASFF member who identifies the risk takes appropriate measures (e.g., a product withdrawal) and transmits the alert. In turn, other members of the system check whether the product in question is on their markets and, if so, also take appropriate measures. Information notifications are used when a risk in food or feed has been identified but other RASFF members do not need to take rapid action because the product has not reached their market or is no longer on their market, or the nature of the risk does not require rapid action. Border rejections may concern products that have been tested and rejected at the external borders of the EEA. Notifications of this type are sent to all other EEA border posts in order to introduce controls and prevent the rejected product from entering the EEA via another border post [9,10].

## 1.2. Products of Plant Origin in the RASFF

Among the product categories reported in the RASFF, the following can be considered as products of plant origin: cereals and bakery products, cocoa and cocoa preparations, coffee and tea, fruits and vegetables, herbs and spices and nuts, nut products and seeds (all product categories that appeared in the RASFF in the period 1979–2020 are shown in Table S1 in the Supplementary Material).

Notifications reported in the RASFF between 1979 and 2020 on products of plant origin are shown in Figure 1. During the period in question, 33,264 notifications were made regarding these products, representing more than 43% of the notifications in the system.

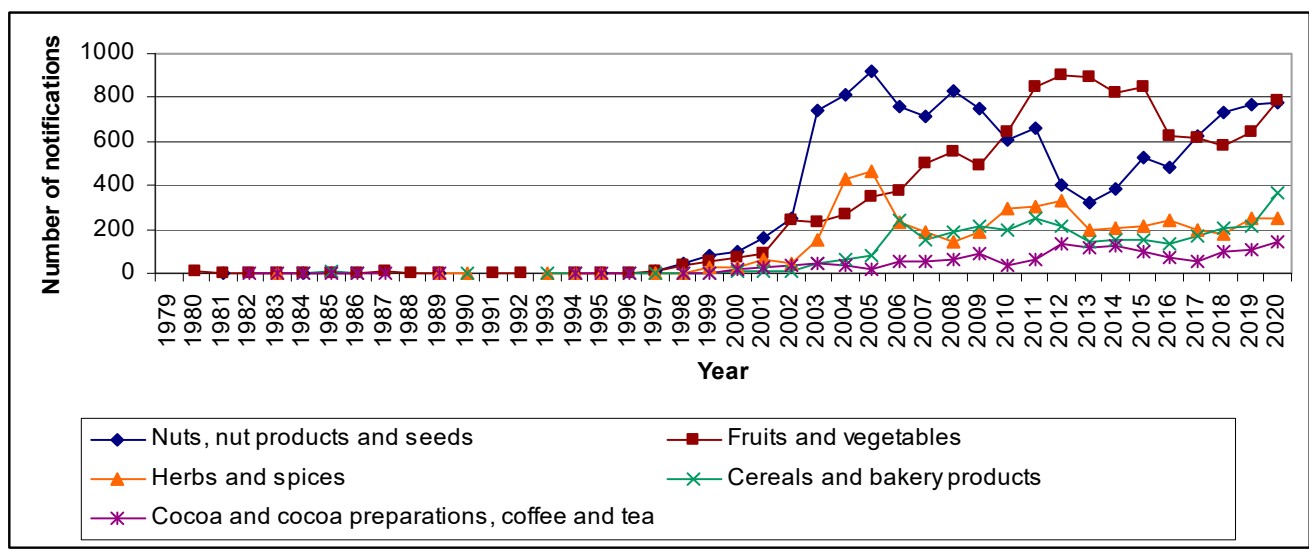

**Figure 1.** Number of notifications for product categories of plant origin in the RASFF in 1979–2020.

The largest number of notifications concerned nuts, nut products and seeds and fruits and vegetables (37% and 35%, respectively, of all notifications to plants in the period 1979–2020). Between 2009 and 2010, a 27% decrease in the number of notifications to nuts, nut products and seeds can be observed, and, in 2009, a 12% decrease in the number of notifications to fruits and vegetables can be seen. This may be related to the introduction of border rejections in the RASFF in 2008. However, in 2010, there was already an increase in the number of notifications for fruits and vegetables, and a slow growth for nuts, nut products and seeds, with around 800 notifications for both categories in 2020.

Annually, the RASFF reports approximately 2000 notifications on products of plant origin, accounting for a significant share of the notifications on all product categories, i.e., 4000–5000 per year (Figure 2).

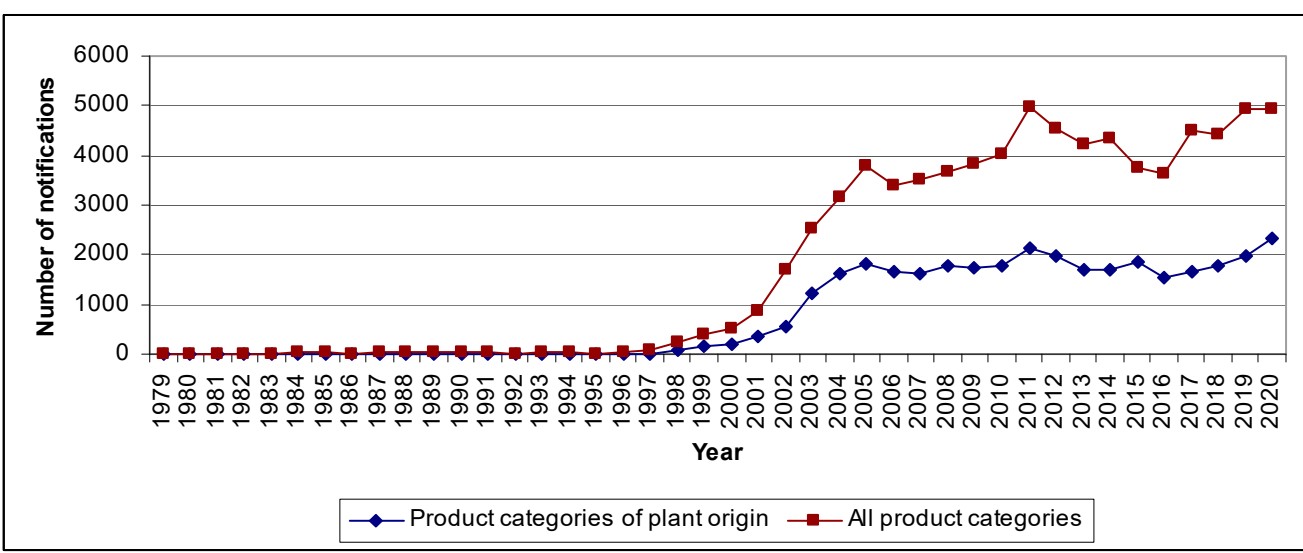

**Figure 2.** Number of notifications on product categories of plant origin and all product categories in the RASFF in 1979–2020.

### 1.3. Goal of the Study

The RASFF's annual reports contain general information about the notifications in this system and mostly only concern the year for which the report was issued. Furthermore, the report for 2021 was even more simplified, and no longer deals with the RASFF alone,

but combines the functioning of the Alert and Cooperation Network (ACN), consisting of three networks: the RASFF, the Administrative Assistance and Cooperation (AAC) and the Agri-Food Fraud Network (FFN) [11].

Other studies relating to notifications on plants in the RASFF mostly concern short periods of a few years. Furthermore, they usually do not indicate individual products or hazards, but only product categories or hazard categories.

Thus, the goal of the study was to examine RASFF notifications for products of plant origin with respect to hazard, year, product, notifying country, origin country, notification type, notification basis, distribution status and action taken in 1998–2020 (23 years).

## 2. Materials and Methods

### 2.1. Data

2.1.1. Hazards Analysed

The data available in the RASFF notifications pre-2021 public information database cover 33,264 notifications (records) on products of plant origin reported in 1980–2020 [8]. This study was limited to the years 1998–2020, during which 33,163 notifications were made. They concerned 582 different hazards reported in 28 categories (Table S2 in the Supplementary Material). All these notifications were subjected to a general analysis (notification summaries in Section 3.1 and joining cluster analysis in Section 3.2).

Meanwhile, hazards with more than 200 notifications (Table 1) were examined in detail (two-way joining cluster analysis in Section 3.3). These were 22 hazards, for which 22,687 notifications were made between 1998 and 2020 (68% of the notifications on plants during this period). Most were notified in one category, but hazards such as colour, *Escherichia coli* and sulphite were reported in two categories, which resulted from the nature of the hazard and its classification by the supervisory authorities.

**Table 1.** The 22 most frequently reported hazards and other hazards regarding food of plant origin notified in the RASFF in 1998–2020.

| Hazard | Number | Hazard Category |
|---|---|---|
| Aflatoxins | 11,465 | Mycotoxins |
| Ochratoxin A | 900 | |
| Ethylene oxide | 463 | Pesticide residues |
| Chlorpyrifos | 437 | |
| Carbendazim | 328 | |
| Dimethoate | 326 | |
| Methomyl | 259 | |
| Acetamiprid | 254 | |
| Omethoate | 222 | |
| Triazophos | 209 | |
| Formetanate | 203 | |
| *Salmonella* | 2584 | Pathogenic micro-organisms |
| *Escherichia coli* | 201 | Microbial contaminants (other) (159), Pathogenic micro-organisms (42) |
| Moulds | 468 | Microbial contaminants (other) |
| Sudan | 790 | Composition |
| Iodine | 219 | |
| Sulphite | 790 | Food additives and flavourings (602), Allergens (188) |
| Colour | 536 | Food additives and flavourings (532), Composition (4) |
| Genetically modified plants | 708 | Genetically modified food or feed |
| Insects | 646 | Foreign bodies |
| Health certificate(s) | 472 | Adulteration/fraud |
| Milk | 207 | Allergens |
| All the above 22 hazards | 22,687 | |
| Other 560 hazards | 10,476 | |
| Total | 33,163 | |

### 2.1.2. Data Processing

The data were processed using Microsoft Excel (Microsoft Corporation, Redmond, DC, USA) applying pivot tables, a vertical search function, filtering and sorting. For product names, sometimes a common name, a Latin name (or in another language, but not in English) or even a proper name was recorded in the database. Different English names referring to the same products were also applied, or information regarding the part of the products (e.g., root, flakes, flower, sprouts), cultivation/breeding method (organic), country/area of origin, the state/degree of processing (e.g., canned, chopped, dried, frozen, milled, roasted, paste), kind/type (e.g., flour, kernels, pickles), the taste (e.g., sweet, bitter) or colour was included. All these entries needed to be standardised to distinguish the basic name, preferably the species name and name related to product category. If a product was classified in the inappropriate product category, this was also corrected.

In the case of products such as peppers and paprika, prunes and plums, the original names were retained (as the products were different), but in other cases the names have been changed to those used in British English (e.g., corn to maize, eggplants to aubergines) and sultanas were changed to raisins and noodles to pasta. In the case of ready-to-use or multi-ingredient products, the final name of the product was given (e.g., cake, bread). If the product consisted of multiple products, was a mixture of products, or could not be identified, the following phrases were used: "(other nuts)", "(other fruits)", "(other vegetables)", "(other herbs)", "(other spices)", "(other seeds)", "(other leaves)" or "(other product)".

Since 2011, information notifications have been divided into information for attention and information for follow-up. In order to standardise this type of notification throughout the research period, these names were shortened to their former name, i.e., information notifications. In the case of variables, such as notification basis, distribution status and action taken, the names of some values in the figures provided in the Supplementary Material were also shortened (by shortening or deleting certain words) to make them easier to handle (Table S3 in the Supplementary Material). In addition, in the case of variables, such as hazard category, notification basis, distribution status and action taken, the empty cells were filled with the phrase "(not specified)".

### 2.1.3. Comments on the RASFF Databases

The data used in this research (i.e., data from the RASFF notifications pre-2021 public information database) came from the archived European Commission website [8]. This is because the present official RASFF database (i.e., RASFF Portal) only contains data from 1.01.2020 onwards, and historical data are solely available to supervisory authorities. Furthermore, the source table exported from this database to a Microsoft Excel file does not contain information on the notification basis, the distribution status, or the action taken [12]. Obtaining information on the hazard category would require exporting data for each such category separately.

### *2.2. Methods*

To identify similarities in the notifications, a cluster analysis was applied using the joining and two-way joining methods. Data were first prepared in tables in Microsoft Excel. The use of these methods required the empty cells to be filled with the value 0. The data were then transferred to Statistica 13.3 (TIBCO Software Inc., Palo Alto, CA, USA).

### 2.2.1. The Joining Cluster Analysis

In order to apply the joining cluster analysis, eight data tables were constructed with all 582 hazards in the rows and the values of each variable in the columns, i.e., year, product, notifying country, origin country, notification type, notification basis, distribution status and action taken. Due to the statistical method used in the case of some hazards, the number of columns with products, origin countries and actions taken was limited to the first 30 with the highest number of notifications. In the joining cluster analysis, the following

settings were used: linkage rule–Ward's method, distance measure–Euclidean measures and vertical icicle plots.

Charts showing the findings of the cluster analysis using the joining method are provided in Supplementary Material in Figure S1 (panels (a)–(h), separately for each mentioned variable).

### 2.2.2. The Two-Way Joining Cluster Analysis

Two-way joining cluster analysis was used when both cases and variables are expected to form clusters simultaneously. Difficulties in interpreting the results may arise because similarities between different clusters may lead to different subsets of variables, so the cluster structure is not homogeneous by nature. However, this method can be considered a powerful data exploration tool [13].

For each of the 22 most frequently reported hazards, seven tables were prepared. The years were placed in the rows and the values of the following variables were noted: product, notifying country, origin country, notification type, notification basis, distribution status and action taken. These were put into columns. Thus, a total of 154 data tables were constructed, with the number of columns for product, origin country and action taken limited to the 30 with the highest number of notifications.

Charts showing the findings of the cluster analysis using the two-way joining method are provided in the Supplementary Material in Figures S2–S23 (for the 22 hazards analysed) in panels (a)–(g), separately for each mentioned variable. These are contour/discrete charts and show the clusters by means of coloured squares (from green, through from yellow, orange and red to brown, where the clusters were highest). The dark green colour was faded (to white) as it would take up the largest part of each chart and would not provide any information.

### 3. Results

#### 3.1. General Results Related to All RASFF Notifications

The overall results considered all 33,163 notifications (based on 582 hazards) reported in the RASFF for products of plant origin in 1998–2020, but were limited to 10 values for particular variables.

#### 3.1.1. Product Categories and Products

Table 2 presents the product categories with 10 most-notified products of plant origin in the RASFF in 1998–2020.

**Table 2.** Product categories with the 10 most-notified products of plant origin in the RASFF in 1998–2020.

| Product Category (Notifications) | Product (Notifications) |
|---|---|
| Cereals and bakery products (3189) | Rice (984), Pasta (334), Maize (242), Biscuits (147), Wheat (109), Cake (91), Bread (86), Breakfast Cereals (80), Buckwheat (73), Linseed (71), Other (972) |
| Cocoa and cocoa preparations, coffee and tea (1490) | Tea (637), Chocolate (298), Coffee (166), Herbal Tea (110), Cocoa (89), Hibiscus (27), Jasmine (21), Senna (20), Fennel (15), Camomile (10), Other (97) |
| Fruits and vegetables (11,462) | Figs (1361), Peppers (1105), Beans (659), Raisins (472), Apricots (460), Betel (431), Okra (406), Mushrooms (342), Grapes (280), Chilli (266), Other (5680) |
| Herbs and spices (4601) | Chilli (779), Curry (547), Pepper (419), Paprika (276), Nutmeg (225), Mint (206), Basil (193), Peppers (168), Coriander (162), Ginger (158), Other (1468) |
| Nuts, nut products and seeds (12,421) | Pistachios (3824), Groundnuts (2281), Peanuts (1586), Sesame (1276), Hazelnuts (1023), Almonds (598), Melons (183), Brazil nuts (155), Rapeseed (150), Pine Nuts (148), Other (1197) |

As previously indicated in Figure 1, the highest number of notifications was reported for nuts (the three most frequently notified products were pistachios, groundnuts and peanuts) and also fruits and vegetables (figs, peppers and beans). Chilli, curry and pepper were mainly reported in the category "herbs and spices", rice, pasta and maize in the category "cereals and bakery products", and tea, chocolate and coffee were notified in the category covering cocoa, coffee and tea. It is also worth noting that peppers and chilli were reported under both the "fruits and vegetables" and "herbs and spices" categories.

### 3.1.2. Other Variables

Table 3 presents the values of the following variables: notifying country, origin country, notification type, notification basis, distribution status and action taken in relation to products of plant origin notified in the RASFF in 1998–2020. The number of these values was limited to the 10 types of notifications that were reported most frequently.

**Table 3.** The notifying country, origin country, notification type, notification basis, distribution status and action taken in notifications on products of plant origin in the RASFF in 1998–2020.

| Variable | Values (Notifications) |
| --- | --- |
| Notifying country | Germany (5735), United Kingdom (3960), Italy (3311), Netherlands (3240), Spain (2030), France (1799), Greece (1500), Poland (1266), Bulgaria (1187), Belgium (1144), other notifying countries (7991) |
| Origin country | Turkey (5137), India (3305), China (3190), Iran (2837), United States (1406), Thailand (1123), Egypt (1024), Netherlands (880), Germany (819), Italy (791), other country of origin (12,651) |
| Notification type | Border rejection (137,25), information (12,986), alert (6452) |
| Notification basis | Border control: consignment detained (17,822), official control on the market (8576), company's own check (2298), (not specified) (1405), border control-consignment released (1270), consumer complaint (851), border control: consignment under customs (633), food poisoning (204), official control in non-member country (45), official control following RASFF notification (44), monitoring of media (15) |
| Distribution status | No distribution (9000), product not (yet) placed on the distribution status market (7525), (not specified) (3533), distribution restricted to notifying country (3418), distribution to other member countries (3210), distribution possible (2677), information on distribution not (yet) available (788), product forwarded to destination (649), product (presumably) no longer on the market (648), product already consumed (623), other distribution status (1092) |
| Action taken | Re-dispatch (7550), destruction (4517), withdrawal from market (3749), official detention (2731), import not authorised (2423), recall from consumers (1767), (not specified) (1357), product recall or withdrawal (1230), return to consignor (783), informing recipient(s) (713), other action taken (6343) |

Products of plant origin were mainly notified by Germany, the United Kingdom, Italy, the Netherlands, and Spain, and originated from outside the European Union (Turkey, India, China, Iran and the United States). Consequently, the most common basis for notification was border control, followed by detention of the consignment and then border rejection. Information notifications and, to a much lesser extent, alerts were also reported. Notifications could also be based on official controls on the market or the company's own checks. Products were most often not distributed or not yet placed on the market, but distribution could also involve the notifying country as well as other member countries. Products were re-dispatched, destroyed or withdrawn from the market.

### 3.2. Results of Joining Cluster Analysis with all RASFF Notifications

In the joining cluster analysis, all 33,163 notifications (based on 582 hazards) were included. In tables prepared for this analysis, rows contained hazards and the columns contained the values of individual variables, i.e., year, product, notifying country, origin country, notification type, notification basis, distribution status and action taken. The number of products, origin countries and actions taken was limited to 30. The results of the joining cluster analysis are shown in the Supplementary Material in Figure S1 in panels (a)-(h) and summarised in Table 4. Next to the individual variable, the most distinct (separated) cluster was indicated first. The pairs of values of a given variable that were

most similar to each other (with regard to the notified hazards) were linked by a long dash, but there were also single-element clusters.

**Table 4.** Results of the joining cluster analysis related to notifications regarding products of plant origin reported in the RASFF in 1998–2020.

| Variable (Figure in Supplemenary Material) | Clusters or Subclusters |
| --- | --- |
| Year (Figure S1a) | First: 2004–2005, 2006–2008, 2007–2010, 2003, 2009<br>Second: 2013–2014, 2015–2018, 2016–2017, 2011, 2012, 2019, 2020<br>Third: 1999–2000, 1998, 2001, 2002 |
| Product (Figure S1b) | First: figs–hazelnuts, peanuts, groundnuts, pistachios<br>Second: pepper–betel, sesame<br>Third: rice–apricots, chilli–almonds<br>Other products |
| Notifying country (Figure S1c) | First: Netherlands–Italy, Spain–France, United Kingdom, Germany, Greece<br>Second: Bulgaria–Belgium, Norway–Finland, Denmark–Czechia, Slovakia–Portugal, Slovenia–Luxemburg, Poland, Sweden, Austria<br>Other notifying countries |
| Origin country (Figure S1d) | First: Iran–Turkey<br>Second: United States–China, Brazil–Egypt, Argentina, India<br>Third: Sudan–Thailand<br>Other origin countries |
| Notification type (Figure S1e) | First: alert<br>Second: information–border rejection |
| Notification basis (Figure S1f) | First: border control: consignment detained<br>Second: official control on the market<br>Third: border control: consignment released–border control: consignment under customs, company's own check, (not specified)<br>Other notification basis– |
| Distribution status (Figure S1g) | First: product not (yet) placed on the market–(not specified), no distribution<br>Second: distribution restricted to notifying country–distribution on the market (possible), distribution to other member countries<br>Other distribution status |
| Action taken (Figure S1h) | First: re-dispatch<br>Second: import not authorised–official detention, withdrawal from market–destruction<br>other action taken |

For the variable year, notifications can be divided into three sub-periods: 2003–2010 (a clear separate cluster), 2011–2020 and 1998–2002. In some cases, pairs of values were formed by consecutive years, meaning that similar hazards were reported at the turn of the year or even for two years (1999 and 2000, 2004 and 2005, 2013 and 2014, 2016 and 2017). Mostly, however, the clusters were formed by years not immediately following each other, meaning that there were fluctuations in the type of hazards reported.

In the case of the variable product, the first cluster was formed by different types of nuts, although the notifications for figs and hazelnuts were the most similar in terms of reported hazards. Notifications for pepper and betel, chilli and almonds, and rice and apricots were also similar.

Considering the notifying countries, the notifications reported by Western European countries, especially the Netherlands and Italy, as well as Spain and France, were the most similar (the United Kingdom was also included in this cluster). This may be indicative of the strong economic links between these countries. The second cluster included medium-sized countries, which were directly paired, e.g., Bulgaria and Belgium, Norway and Finland, Denmark and Czechia.

In the case of countries of origin, a distinct cluster was formed by Iran and Turkey. It is reasonable to assume that the number of notifications regarding the hazards originating from these countries was high. It is important to remark that, in the case of the variable no-

tification basis, a one-element cluster "border control-consignment detained" was formed, with a significant linkage distance from the other values of this variable.

It is also worth noting that the second cluster of the variable origin country did not include EU countries. This means that most of the hazards regarding plant products came from non-EU countries. Considering the variable notification type, it can be seen that border rejections were more similar to information notifications than to alert notifications. Thanks to the border rejections, the border posts of the EU countries contributed, to a large extent, to the minimisation of hazards in products. This was also confirmed by the values of other variables. Indeed, considering the variable distribution status, the first cluster was formed by the values: "product not (yet) placed on the market" and "no distribution". However, in the case where a variable action was taken, the first one-element cluster was created by the value "re-dispatch", and in the second cluster, similar values were "import not authorised" and "official detention".

However, it is also important to note the other values of the individual variables, which can also be linked in a sequence concerning products of plant origin that are already on the EU market. In the case of the variable notification basis, the second cluster was formed by the value "official control on the market". When considering the variable distribution status, a similarity can be seen between the values: "distribution restricted to notifying country" and "distribution on the market (possible)" (second cluster). In turn, in the case of the variable action taken, a similarity can be observed between the values "withdrawal from market" and "destruction" (also second cluster).

### 3.3. Results of Two-Way Joining Cluster Analysis with Selected RASFF Notifications

The selected 22 hazards (reported under 22,687 notifications) indicated in Table 1 were considered for the two-way joining cluster analysis. The results of this analysis are presented in Figures S2–S23 in the Supplementary Material, where panels (a)–(g) show the similarity between year and product, notifying country, origin country, notification type, notification basis, distribution status and action taken, respectively. Based on the individual years of the variable year, the values of the other variables were indicated, with the product as the base variable, i.e., panel (a). If there was no coverage of the same years in the other variables, they were omitted. In some cases, the variation in cluster intensity (dependent on colours) in particular years caused the name of the product to be determined by the values of the other variables. This made it possible to focus only on the most distinct clusters that occurred simultaneously in the different variables.

In Sections 3.3.1–3.3.9, the hazards reported in particular categories are presented.

### 3.3.1. Mycotoxins (*Aflatoxins* and Ochratoxin A)

Notifications relating to mycotoxins (aflatoxins and ochratoxin A) are presented in Table 5. These notifications were reported most frequently and accounted for up to 55% of the notifications examined using two-way joining cluster analysis. They mainly concerned products from Asia, but aflatoxins in pistachios from Iran were the most prominent problem. This hazard was particularly prevalent between 2003 and 2006, and was reported by Germany and Spain using information notifications after border control. Consignments were detained and re-dispatched, resulting in products not being distributed.

Ochratoxin A in raisins from Turkey was notified in 2016–2019. These products were reported by Germany, the Netherlands, Poland and France, at both border and official controls at the market. They were withdrawn from the market, destroyed or dispatched.

**Table 5.** Results of the two-way joining cluster analysis related to notifications regarding mycotoxins in plants reported in the RASFF in 1998–2020.

| | Hazard/Variable | Value (Figure in Supplementary Material) |
|---|---|---|
| **Aflatoxins** | Year | 2003–2006 |
| | Product | Pistachios (2003–2006) (Figure S2a) |
| | Notifying country | Germany (2003–2006), Spain (2004, 2005) (Figure S2b) |
| | Origin country | Iran (2003–2006) (Figure S2c) |
| | Notification type | Information (2003–2006) (Figure S2d) |
| | Notification basis | Border control: consignment detained (2003–2006) (Figure S2e) |
| | Distribution status | (Not specified) (2003, 2004), no distribution (2005, 2006) (Figure S2f) |
| | Action taken | Re-dispatch (2003–2006) (Figure S2g) |
| **Ochratoxin A** | Year | 2006, 2016, 2018, 2019 (for some variables, there was not full coverage in years) |
| | Product | Raisins (2006, 2016, 2018, 2019) (Figure S3a) |
| | Notifying country | Czechia, Italy (2006), Germany (2016, 2018), Netherlands (2016, 2018, 2019), Poland (2018, 2019), France (2019) (Figure S3b) |
| | Origin country | Turkey (2018, 2019) (Figure S3c) |
| | Notification type | Information (2006), alert (2016, 2018, 2019), border rejections (2018, 2019) (Figure S3d) |
| | Notification basis | Official control on the market (2006, 2016, 2018, 2019), border control: consignment detained (2018, 2019) (Figure S3e) |
| | Distribution status | Distribution on the market (possible) (2006), product not (yet) placed on the market (2016, 2018, 2019) (Figure S3f) |
| | Action taken | Product recall or withdrawal (2006), re-dispatch (2006, 2018, 2019), destruction, informing recipient(s) (2016), re-dispatch, withdrawal from the market (2016, 2018, 2019), official detention, return to consignor (2019) (Figure S3g) |

### 3.3.2. Pesticide Residues (Ethylene oxide, Chlorpyrifos, Carbendazim, Dimethoate, Methomyl, Acetamiprid, Omethoate, Triazophos and Formetanate)

Notifications relating to pesticide residues are presented in Table 6. This was the largest group of reported hazards (9 different substances out of the 22 analysed hazards).

Products with these hazards usually originated from Asia. Of particular note is the presence of pesticide residues in peppers from Turkey, as notified by Bulgaria in several years. These included substances such as acetamiprid in 2020, chlorpyrifos in 2016, 2017 and 2019, formetanate in 2011, 2012, 2014 and 2017–2020 and methomyl in 2010, 2011 and 2018. The type of notification was border rejection based on border controls, after which the consignment was detained. Products were, therefore, not placed on the market or not distributed, and were most often destroyed thereafter.

Another country that frequently appeared in notifications relating to pesticide residues was India. France and the United Kingdom reported acetamiprid, dimethoate and triazophos in okra in 2012 and 2013 and triazophos in curry in the same years. Italy notified carbendazim in rice in 2014 and 2015. There were border rejections based on border controls, followed by detention of the consignment. The products were, therefore, not distributed and usually were destroyed. There was a more serious problem with ethylene oxide, which was notified by the Netherlands in sesame in 2020. Products with this hazard were reported as alerts after the companies' own checks, so distribution to other EU countries was possible. Actions such as informing consignors and recipients, recalls and withdrawals were then taken.

Acetamiprid was also notified in 2012 and 2013 by France and the United Kingdom in tea from China and India, and carbendazim was reported in 2010 by the United Kingdom in peppers from Thailand. Notifications related to dimethoate in beans and peas from Egypt and Kenya, respectively, were sent in 2013 by France. Many countries also reported the presence of omethoate in beans, aubergines, apples, okra and peppers from Thailand in 2006, 2008–2013 and 2019.

**Table 6.** Results of the two-way joining cluster analysis related to notifications on pesticide residues in plants reported in the RASFF in 1998–2020.

| | Hazard/Variable | Value (Figure in Supplementary Material) |
|---|---|---|
| **Ethylene oxide** | Year | 2020 (this year occurred for each value of each variable below) |
| | Product | Sesame (Figure S4a) |
| | Notifying country | Netherlands (Figure S4b) |
| | Origin country | India (Figure S4c) |
| | Notification type | Alert (Figure S4d) |
| | Notification basis | Company's own check (Figure S4e) |
| | Distribution status | Distribution to other member countries (Figure S4f) |
| | Action taken | Informing consignor, informing recipient(s), recall from consumers, withdrawal from the market (Figure S4g) |
| **Chlorpyrifos** | Year | 2016, 2017, 2019 (these years occurred for each value of each variable below) |
| | Product | Peppers (Figure S5a) |
| | Notifying country | Bulgaria (Figure S5b) |
| | Origin country | Turkey (Figure S5c) |
| | Notification type | Border rejections (Figure S5d) |
| | Notification basis | Border control: consignment detained (Figure S5e) |
| | Distribution status | Product not (yet) placed on the market (Figure S5f) |
| | Action taken | Destruction (Figure S5g) |
| **Carbendazim** | Year | 2010, 2014, 2015 |
| | Product | Peppers (2010), rice (2014, 2015) (Figure S6a) |
| | Notifying country | United Kingdom (2010), Italy (2014, 2015) (Figure S6b) |
| | Origin country | Thailand (2010), India (2014, 2015) (Figure S6c) |
| | Notification type | Information (2010), border rejection (2014, 2015) (Figure S6d) |
| | Notification basis | Official control on the market (2010), border control: consignment detained (2010, 2014, 2015), Border control: consignment under customs (2015) (Figure S6e) |
| | Distribution status | No distribution (2010), product not (yet) placed on the market (2014, 2015), product forwarded to distribution (2015) (Figure S6f) |
| | Action taken | Destruction (2010, 2014, 2015), withdrawal from market (2010, 2014), re-dispatch (2014, 2015) (Figure S6g) |
| **Dimethoate** | Year | 2012, 2013 |
| | Product | Okra (2012), beans, okra, peas (2013) (Figure S7a) |
| | Notifying country | United Kingdom (2012, 2013), France (2013) (Figure S7b) |
| | Origin country | India (2012), Egypt, India, Kenya (2013) (Figure S7c) |
| | Notification type | Border rejection (2012, 2013) (Figure S7d) |
| | Notification basis | Border control: consignment detained (2012, 2013) (Figure S7e) |
| | Distribution status | No distribution (2012), product not (yet) placed on the market (2013) (Figure S7f) |
| | Action taken | Destruction (2012, 2013) (Figure S7g) |
| **Methomyl** | Year | 2010, 2011, 2018 |
| | Product | Peppers (2010, 2011, 2018) (Figure S8a) |
| | Notifying country | Bulgaria (2010, 2011, 2018) (Figure S8b) |
| | Origin country | Turkey (2010, 2011, 2018) (Figure S8c) |
| | Notification type | Border rejections (2010, 2011, 2018) (Figure S8d) |
| | Notification basis | Border control: consignment detained (2010, 2011, 2018) (Figure S8e) |
| | Distribution status | No distribution (2010, 2011), product not (yet) placed on the market (2018) (Figure S8f) |
| | Action taken | Destruction (2010, 2011, 2018) (Figure S8g) |
| **Acetamiprid** | Year | 2012, 2013, 2020 |
| | Product | Tea (2012), tea, okra (2013), peppers (2020) (Figure S9a) |
| | Notifying country | France (2012), France, United Kingdom (2013), Bulgaria (2020) (Figure S9b) |
| | Origin country | China, India (2012, 2013), Turkey (2020) (Figure S9c) |
| | Notification type | Border rejection (2012, 2013, 2020) (Figure S9d) |
| | Notification basis | Border control: consignment detained (2012, 2013, 2020) (Figure S9e) |
| | Distribution status | No distribution (2012), product not (yet) placed on the market (2013, 2020) (Figure S9f) |
| | Action taken | Destruction (2012, 2020), import not authorised (2013) (Figure S9g) |

**Table 6.** *Cont.*

| | Hazard/Variable | Value (Figure in Supplementary Material) |
|---|---|---|
| **Omethoate** | Year | 2006, 2008–2013, 2019 (for some variables, there was not full coverage in years) |
| | Product | Beans (2006, 2008, 2011), aubergines (2009, 2010, 2012, 2013), apples (2010), okra (2013), peppers (2019) (Figure S10a) |
| | Notifying country | Norway (2006), Netherlands (2008, 2010, 2011), Finland (2009), Germany (2010, 2012), France (2013), United Kingdom (2013, 2019), Belgium, Bulgaria (2018) (Figure S10b) |
| | Origin country | Thailand (2006, 2008–2010) (Figure S10c) |
| | Notification type | Information (2006, 2008–2013, 2019), border rejection (2009–2013, 2019) (Figure S10d) |
| | Notification basis | Official detention (2006, 2008, 2010, 2011), border control: consignment detained (2009–2013, 2019) (Figure S10e) |
| | Distribution status | No distribution (2009–2012), product already consumed (2012), product not (yet) placed on the market (2013, 2019) (Figure S10f) |
| | Action taken | Withdrawal from the market (2009, 2011), destruction (2009, 2010, 2013, 2019), informing authorities (2012, 2013) (Figure S10g) |
| **Triazophos** | Year | 2012, 2013 |
| | Product | Curry (2012), okra (2012, 2013) (Figure S11a) |
| | Notifying country | France (2012), United Kingdom (2012, 2013) (Figure S11b) |
| | Origin country | India (2012, 2013) (Figure S11c) |
| | Notification type | Border rejection (2012, 2013) (Figure S11d) |
| | Notification basis | Border control: consignment detained (2012, 2013) (Figure S11e) |
| | Distribution status | No distribution (2012, 2013) (Figure S11f) |
| | Action taken | Destruction (2012, 2013) (Figure S11g) |
| **Formetanate** | Year | 2011, 2012, 2014, 2017–2020 |
| | Product | Peppers (2011, 2012, 2014, 2017–2020) (Figure S12a) |
| | Notifying country | Bulgaria (2011, 2012, 2014, 2017–2020) (Figure S12b) |
| | Origin country | Turkey (2011, 2012, 2014, 2017–2020) (Figure S12c) |
| | Notification type | Border rejection (2011, 2012, 2014, 2017–2020) (Figure S12d) |
| | Notification basis | Border control: consignment detained (2011, 2012, 2014, 2017–2020) (Figure S12e) |
| | Distribution status | No distribution (2011, 2012), product not (yet) placed on the market (2014, 2017–2020) (Figure S12f) |
| | Action taken | Re-dispatch or destruction (2011), placed under customs seals (2012), destruction (2017–2020) (Figure S12g) |

### 3.3.3. Pathogenic Micro-Organisms and Microbial Contaminants (*Salmonella*, *Escherichia coli* and Moulds)

Notifications regarding pathogenic micro-organisms are presented in Table 7. Hazards related to *Salmonella* presence have been reported in recent years (2015, 2018 and 2019) by the United Kingdom, Greece and Germany in sesame from India, Sudan and Brazil. Notifications were reported as border rejections on the basis of controls, after which shipments were detained. Consequently, the products were not placed on the market and were destroyed, re-dispatched or physically/chemically treated.

*Escherichia coli* was reported by Norway and the United Kingdom in 2005, 2012, 2013, 2016 and 2020 in basil, mint and betel from Asian countries, i.e., Thailand, Vietnam and the Lao Republic. These were information notifications sent after official controls on the market or border controls, after which consignment was detained. Distribution was limited to the notifying country or the product was removed from the market. The trade of these products was prohibited, and they were also withdrawn from the market and destroyed.

Mould has been reported over a wide range of time (2007, 2008, 2011, 2012, 2017 and 2018), mainly by Poland in nuts (peanuts, hazelnuts, groundnuts), raisins and beans from Turkey and China. These were information notifications or border rejections, after which the shipments were detained. Consequently, the products were not distributed or had not yet been placed on the market, and were most often re-dispatched.

**Table 7.** Results of the two-way joining cluster analysis related to notifications regarding pathogenic micro-organisms and microbial contaminants in plants reported in the RASFF in 1998–2020.

| | Hazard/Variable | Value (Figure in Supplementary Material) |
|---|---|---|
| *Salmonella* | Year | 2015, 2018,2019 |
| | Product | Sesame (2015, 2018–2020) (Figure S13a) |
| | Notifying country | United Kingdom (2015), Greece (2018, 2019), Germany (2019, 2020) (Figure S13b) |
| | Origin country | India (2015), Sudan (2018, 2019), Brazil (2019, 2020) (Figure S13c) |
| | Notification type | Border rejection (2015, 2018–2020) (Figure S13d) |
| | Notification basis | Border control: consignment detained (2015, 2018–2020) (Figure S13e) |
| | Distribution status | Product not (yet) placed on the market (2015, 2018–2020) (Figure S13f) |
| | Action taken | Destruction (2015), re-dispatch (2015, 2018, 2019), physical/chemical treatment (2019, 2020), official detention (2020) (Figure S13g) |
| *Escherichia coli* | Year | 2005, 2012, 2013, 2016, 2020 |
| | Product | Basil (2005, 2012, 2013, 2016), mint (2005), betel (2020) (Figure S14a) |
| | Notifying country | Norway (2005, 2012, 2013), United Kingdom (2016, 2020) (Figure S14b) |
| | Origin country | Thailand (2005), Vietnam (2012, 2013, 2020), Lao Republic (2016) (Figure S14c) |
| | Notification type | Information (2005, 2012, 2013, 2016, 2020) (Figure S14d) |
| | Notification basis | Official control on the market (2005, 2012, 2013), border control: consignment detained (2016, 2020) (Figure S14e) |
| | Distribution status | Distribution restricted to notifying country (2005, 2012, 2013), product (presumably) no longer on the market (2016) (Figure S14f) |
| | Action taken | Prohibition to trade (2005), withdrawal from the market (2012, 2013, 2016, 2020), destruction (2016) (Figure S14g) |
| Moulds | Year | 2007, 2008, 2011, 2012, 2017, 2018 (for some variables there was not full coverage in years) |
| | Product | Peanuts (2007), beans (2008, 2011), hazelnuts, raisins (2012), groundnuts (2017, 2018) (Figure S15a) |
| | Notifying country | Poland (2007, 2008, 2011) (Figure S15b) |
| | Origin country | China (2007, 2008, 2011), Turkey (2012) (Figure S15c) |
| | Notification type | Information (2007, 2012), border rejection (2008, 2011, 2012, 2017, 2018) (Figure S15d) |
| | Notification basis | Border control: consignment detained (2007, 2008, 2011, 2012, 2017, 2018) (Figure S15e) |
| | Distribution status | No distribution (2007, 2008, 2011, 2012), product not (yet) placed on the market (2017, 2018) (Figure S15f) |
| | Action taken | Re-dispatch (2007, 2008, 2012), return to consignor (2011), withdrawal from the market (2012) (Figure S15g) |

### 3.3.4. Composition (Sudan and Iodine)

Problems regarding composition (Table 8) were reported in products originating from Asia and Europe.

**Table 8.** Results of the two-way joining cluster analysis related to notifications on composition in plants reported in the RASFF in 1998–2020.

| | Hazard/Variable | Value (Figure in Supplementary Material) |
|---|---|---|
| Sudan | Year | 2004, 2005 |
| | Product | (Other spices) (2004, 2005), chilli (2005) (Figure S16a) |
| | Notifying country | Germany (2004, 2005) (Figure S16b) |
| | Origin country | Germany, Italy, Turkey (2004), India (2004, 2005) (Figure S16c) |
| | Notification type | Information, alert (2004, 2005) (Figure S16d) |
| | Notification basis | Official control on the market (2004, 2005) (Figure S16e) |
| | Distribution status | (Not specified) (2004), distribution on the market (possible) (2004, 2005) (Figure S16f) |
| | Action taken | Destruction (2004), product recall or withdrawal (2004, 2005) (Figure S16g) |
| Iodine | Year | 2004, 2005, 2008–2010, 2014, 2018, 2019 (for some variables, there was not full coverage in years) |
| | Product | Seaweed (2004, 2005, 2008–2010, 2014, 2018, 2019), algae (2004, 2005) (Figure S17a) |
| | Notifying country | Germany (2004, 2005, 2008–2010, 2014) (Figure S17b) |
| | Origin country | South Korea (2004, 2005, 2009, 2014, 2018), Netherlands (2004, 2009), China (2004, 2005, 2008, 2010, 2018, 2019) (Figure S17c) |
| | Notification type | Alert (2004, 2005, 2009, 2010, 2014, 2018, 2019), information (2008) (Figure S17d) |
| | Notification basis | Official control on the market (2004, 2005, 2008–2010, 2014, 2018, 2019) (Figure S17e) |
| | Distribution status | (Not specified) (2004), distribution on the market (possible) (2004, 2005, 2008–2010), distribution restricted to notifying country (2008), distribution to other member countries (2018) (Figure S17f) |
| | Action taken | Product recall or withdrawal (2004, 2005), destruction (2008), withdrawal from the market (2009, 2010, 2014, 2018) (Figure S17g) |

Sudan dye was notified by Germany in earlier years (2004 and 2005) in chilli and other spices from Germany, Italy, Turkey and India. These notifications were in the form of information or alert notifications based on official controls on the market, and the reported products were destroyed or withdrawn.

Iodine was reported mainly by Germany in 2004, 2005, 2008–2010, 2014, 2018 and 2019. Algae was notified only in the earlier years (2004 and 2005), while seaweed was submitted in all the mentioned years. The reported products were from South Korea, China and the Netherlands. These were mainly alert notifications and, to a lesser extent, information notifications, sent on the basis of official controls on the market. Distribution status varied widely, with products being destroyed or withdrawn from the market.

### 3.3.5. Food Additives and Flavourings (Sulphite and Colour)

Sulphites and colours (Table 9) were mainly notified in food additives and flavourings category; however, sulphites were also reported as allergens, and colours were also notified within each composition category.

**Table 9.** Results of the two-way joining cluster analysis related to notifications regarding food additives and flavourings in plants reported in the RASFF in 1998–2020.

| | Hazard/Variable | Value (Figure in Supplementary Material) |
|---|---|---|
| **Sulphite** | Year | 2003, 2005, 2014–2018 (for some variables, there was not full coverage in years) |
| | Product | Apricots (2003, 2005, 2014–2018) (Figure S18a) |
| | Notifying country | Spain (2003), Cyprus (2005) (Figure S18b) |
| | Origin country | Turkey (2003, 2005, 2014–2018) (Figure S18c) |
| | Notification type | Information (2003, 2005), alert (2005), border rejection (2014–2018) (Figure S18d) |
| | Notification basis | Border control: consignment detained (2003, 2014–2018), official control on the market (2005) (Figure S18e) |
| | Distribution status | (Not specified) (2003), distribution on the market (possible) (2005), product not (yet) placed on the market (2014–2018) (Figure S18f) |
| | Action taken | Re-dispatch (2003), product recall or withdrawal (2005), import not authorised (2017), recall from consumers (2018) (Figure S18g) |
| **Colour** | Year | 2020 (this year occurred for each value of each variable below) |
| | Product | Breakfast cereals (Figure S19a) |
| | Notifying country | United Kingdom (Figure S19b) |
| | Origin country | United States (Figure S19c) |
| | Notification type | Border rejection (Figure S19d) |
| | Notification basis | Border control: consignment detained (Figure S19e) |
| | Distribution status | Product not (yet) placed on the market (Figure S19f) |
| | Action taken | Official detention (Figure S19g) |

Sulphites in apricots from Turkey were reported both in earlier years (2003 and 2005 as information or alert notifications) and more recently (2014–2018 as border rejections). These notifications were sent by Spain and Cyprus on the basis of an official control on the market or a border control, after which the consignment was detained. The notified products were dispatched and, if found on the market, were withdrawn or recalled from consumers.

Hazards regarding colour were reported by the United Kingdom in 2020 on breakfast cereals originating from the United States. These were border rejections based on border controls, after which the consignment was detained. The products were not (yet) placed on the market, because they were officially detained.

### 3.3.6. Genetically Modified Food

Alerts regarding genetically modified products were raised in 2006 by Austria regarding linseed originating from the United States. In turn, Germany reported this hazard in 2009 in rice from Canada as an information notification. These products were reported on the basis of official control and were withdrawn from the market (Table 10).

**Table 10.** Results of the two-way joining cluster analysis related to notifications regarding genetically modified plants reported in the RASFF in 1998–2020.

| | Hazard/Variable | Value (Figure in Supplementary Material) |
|---|---|---|
| Genetically modified | Year | 2006, 2009 |
| | Product | Linseed (2006), rice (2009) (Figure S20a) |
| | Notifying country | Austria (2006), Germany (2009) (Figure S20b) |
| | Origin country | United States (2006), Canada (2009) (Figure S20c) |
| | Notification type | Alert (2006), information (2009) (Figure S20d) |
| | Notification basis | Official control on the market (2006, 2009) (Figure S20e) |
| | Distribution status | Distribution on the market (possible) (2006, 2009) (Figure S20f) |
| | Action taken | Product recall or withdrawal (2006), withdrawal from the market (2006, 2009) (Figure S20g) |

### 3.3.7. Foreign Bodies (Insects)

Insects (as foreign bodies) were reported mainly in 2006–2009, 2011, 2012 and 2017 (Table 11).

**Table 11.** Results of the two-way joining cluster analysis related to notifications on foreign bodies (insects) in plants reported in the RASFF in 1998–2020.

| | Hazard/Variable | Value (Figure in Supplementary Material) |
|---|---|---|
| Insects | Year | 2006–2009, 2011, 2012, 2017 |
| | Product | Rice (2006, 2011), dates (2007, 2008, 2014), figs (2007, 2008, 2011), almonds (2009, 2011), tea (2009), chocolate (2011), rapeseed (2012), peanuts (2007) (Figure S21a) |
| | Notifying country | Poland (2006–2009, 2011, 2012, 2014), Slovenia (2008), Spain (2009), Italy (2011, 2012), Czechia (2012) (Figure S21b) |
| | Origin country | Turkey (2006–2008), China (2007, 2009), Italy (2007, 2008), United States (2009), Ukraine (2011, 2012), India, Tunisia (2014) (Figure S21c) |
| | Notification type | Information (2006–2008, 2011, 2012), border rejection (2008, 2009, 2011, 2012, 2014) (Figure S21d) |
| | Notification basis | Border control: consignment detained (2006–2009, 2011, 2012, 2014), official control on the market (2006–2008), consumer complaint (2008, 2011) (Figure S21e) |
| | Distribution status | No distribution (2006–2009, 2011, 2012), distribution on the market (possible) (2008, 2009), information on the product not (yet) available (2011), product not (yet) placed on the market (2014) (Figure S21f) |
| | Action taken | Re-dispatch (2006–2009, 2011, 2012, 2017), withdrawal from the market (2007, 2008, 2011, 2012) (Figure S21g) |

These notifications concerned products such as rice, dates, figs, almonds, tea, chocolate, rapeseed and peanuts originated from Asian countries (Turkey, China, India), European countries (Italy and Ukraine), and also the United Stated and Tunisia. They were sent by Poland, Slovenia, Spain, Italy and Czechia as information notifications or border rejections. The notification basis was official control on the market or border control, after which the consignment was detained, as well as consumer complaints. The distribution status of notified products was very diverse, and they were withdrawn from the market or re-dispatched.

### 3.3.8. Adulteration/Fraud (Health Certificate(s))

Problems with health certificates were the cause of notifications within the adulteration/fraud category (Table 12). The notifications concerned products such as nutmeg (in 2016) and chilli, sesame and pistachios (in 2017) from India, reported by the United Kingdom. These were border rejections on the basis of border controls, after which the consignment was detained. Consequently, the products were not placed on the market and were destroyed.



**Table 12.** Results of the two-way joining cluster analysis related to notifications regarding adulteration/fraud (health certificate(s)) in plants reported in the RASFF in 1998–2020.

| | Hazard/Variable | Value (Figure in Supplementary Material) |
|---|---|---|
| Health certificate(s) | Year | 2016, 2017 |
| | Product | Nutmeg (2016), chilli, sesame, pistachios (2017) (Figure S22a) |
| | Notifying country | United Kingdom (2016, 2017) (Figure S22b) |
| | Origin country | India (2016, 2017) (Figure S22c) |
| | Notification type | Border rejection (2016, 2017) (Figure S22d) |
| | Notification basis | Border control: consignment detained (2016, 2017) (Figure S22e) |
| | Distribution status | Product not (yet) placed on the market (2016, 2017) (Figure S22f) |
| | Action taken | Destruction (2016, 2017) (Figure S22g) |

### 3.3.9. Allergens (Milk)

Milk as an allergen (Table 13) in chocolate originating from Germany was reported primarily by Austria in 2009, using alert notifications. These notifications were based on the official controls on the market, and the action taken was to issue a public warning.

**Table 13.** Results of the two-way joining cluster analysis related to notifications regarding allergens (milk) in plants reported in the RASFF in 1998–2020.

| | Hazard/Variable | Value (Figure in Supplementary Material) |
|---|---|---|
| Milk | Year | 2009 (this year occurred for each value of each variable below) |
| | Product | Chocolate (Figure S23a) |
| | Notifying country | Austria (Figure S23b) |
| | Origin country | Germany (Figure S23c) |
| | Notification type | Alert (Figure S23d) |
| | Notification basis | Official control on the market (Figure S23e) |
| | Distribution status | Distribution on the market (possible) (Figure S23f) |
| | Action taken | Public warning: press release (Figure S23g) |

### 3.4. Limitations of Using RASFF Data

The research used data from the archived RASFF database, covering notifications up to 2020 at the time of data extraction [8]. At present, 2021 is also available in this database. A study that also covers the year 2022 would require the data from this database to be combined with the data from the database currently available on the European Commission website [10]. However, this would be difficult due to their different structure, especially once exported to an Excel file. It is also unknown if and when the Commission will officially make the historical data available. At present, they are only available to the supervisory authorities of the member countries. It is also worth mentioning that the current database is much less accessible to the user than the one made officially available a few years ago.

The actual number of notifications placed in the RASFF database was about 20% less than the number of records, as one notification could include several records (concerning, for example, the different countries of origin of the notified product). However, combining the records into a single notification would lead to the loss of a large amount of data, as it would require the adoption of the principle that only the value from one (e.g., the first) record of a notification can be taken into account. Indeed, only one value could occur in each notification within a given variable for further analysis. However, it should be noted that the inclusion of all records allowed for proportionality, and so should not significantly affect the final results.

In the earlier years of the RASFF functioning (1980s and 1990s), missing data could be observed for the variables of hazard category, notification basis, distribution status and action taken (empty cells were filled with the phrase "(not specified)"). It should be added, however, that, due to the small number of notifications in that period, these years were excluded from the study. A major difficulty was the wide variety of product names, as these

were given with their characteristics or states or under different English names. The inability to clearly identify the product, or to only identify the few notifications regarding little-known products, required the creation of new names: "(other fruits)", "(other vegetables)", "(other herbs)", "(other spices)", "(other nuts)", "(other seeds)", "(other leaves)" and "(other product)". Differentiated products were thus concentrated under the same group name. However, this applied to only 3% of the total examined population, and was dispersed across the five studied product categories.

In the source tables prepared for the cluster analysis in Statistica 13.3, a maximum of approximately thirty columns (for the joining method) and, similarly, a maximum of approximately thirty columns and thirty rows (for the two-way joining method) could be included. A larger number of columns and/or rows could significantly impair the readability of the charts generated based on these. Therefore, sorting was carried out from the largest to the smallest sum of values (up to the aforementioned number of about thirty), and the others were omitted. However, this allowed for us to focus on the most significant clusters. In turn, the use of Ward's method as a linkage rule in joining cluster analysis enabled a good separation of clusters, but caused them to be flattened (this is, however, a characteristic of this method), which sometimes made it difficult to read the charts.

Difficulties also arose from the use of the two-way joining cluster analysis method. Although each variable (i.e., product, notifying country, origin country, notification type, notification basis, distribution status and action taken) consecutively referred to the same variable, i.e., year, it was sometimes possible to see values concentrating (clustering) within one variable and dispersing within another variable. This caused difficulties in interpretation, as, for some variables, there was not full clustering coverage within the same years. In addition, when generating the charts, the Statistica program did not accurately map the colours from the legend to the colours on the chart and omitted every second mark (on each axis), resulting in the need to manually modify each chart. Furthermore, the clusters were not arranged according to consecutive years, but according to the number of notifications in different years (so there was no continuity over time in the charts). As a final difficulty, the charts were automatically rescaled in such a way that they did not take up all the available space (much of the space was left blank). This caused the graphical and textual elements of the charts to be reduced in size, and thus compromised its readability.

## 4. Discussion

### 4.1. The Annual RASFF Reports

Notifications regarding hazards in products of plant origin occurred each year among the so-called "Top 10" included in the annual RASFF reports for 2010–2020. They covered information on hazard, product category, origin country and notifying country (Table 14). In reports for earlier years, information on the "Top 10" was not provided. It is also worth mentioning that, in the RASFF annual reports, information on the notification basis, distribution status and action taken is not given in "Top 10", but only within a selected case study for a particular product in a given year.

Of particular note is the indication of aflatoxins in nuts (mainly from China, Iran and Turkey, and notified by Germany, Italy, the Netherlands and the United Kingdom) in each of the 2010–2020 RASFF annual reports. In turn, according to the results of the two-way joining cluster analysis presented in Section 3.3.1, this hazard was reported on nuts from Iran between 2003 and 2006. There may be two reasons for this difference: firstly, the "Top 10" summaries were not included in earlier RASFF annual reports (i.e., for years prior to 2010), and secondly, the number of notifications for nuts between 2003 and 2006 was so high that the cluster analysis showed it to be the highest concentration, omitting the subsequent years of the analysed period. It should be noted, however, that this hazard is an ongoing, significant problem signalled in the annual reports, despite its noticeable reduction. Importantly, aflatoxins were also reported almost every year in fruits and vegetables from Turkey, and in 2018 and 2019 this was also related to ochratoxin A (which coincides with the results of the cluster analysis).

Table 14. Hazards in products of plant origin in the annual RASFF reports for 2010–2020.

| Year | Hazard | Product Category | Origin Country * | Notifying Country * | Reference |
|---|---|---|---|---|---|
| 2010 | Aflatoxins | Fruits and vegetables Herbs and spices | Turkey India | NDA United Kingdom | [14] |
| | | Nuts, nut products and seeds | Argentina, China, Iran, Turkey, United States | Germany, Greece, Italy, The Netherlands, Spain, United Kingdom | |
| | Unauthorized genetically modified | Cereals and bakery products | China | NDA | |
| 2011 | Aflatoxins | Fruits and vegetables Herbs and spices | Turkey India | NDA NDA | [15] |
| | | Nuts, nut products and seeds | China, Turkey, Iran | Germany, The Netherlands, United Kingdom | |
| | *Salmonella* | Fruits and vegetables | Bangladesh | United Kingdom | |
| | Living and died mites | Nuts, nut products and seeds | Ukraine | Poland | |
| 2012 | Aflatoxins | Fruits and vegetables | Turkey | France | [16] |
| | | Nuts, nut products and seeds | China | Germany, The Netherlands, United Kingdom | |
| | Monocrotophos | Fruits and vegetables | India | NDA | |
| | *Salmonella* | Fruits and vegetables | Bangladesh | United Kingdom | |
| 2013 | Aflatoxins | Fruits and vegetables | Turkey | NDA | [17] |
| | | Nuts, nut products and seeds | China, Turkey | Germany, Italy, The Netherlands | |
| 2014 | Aflatoxins | Fruits and vegetables | Turkey | NDA | [18] |
| | | Nuts, nut products and seeds | China, Iran, Turkey | Germany, Italy, The Netherlands, United Kingdom | |
| | Dichlorvos | Fruits and vegetables | Nigeria | United Kingdom | |
| 2015 | Aflatoxins | Fruits and vegetables | Turkey | NDA | [19] |
| | | Nuts, nut products and seeds | China, Iran, Turkey, United States | Belgium, Germany, Italy, The Netherlands, Spain, United Kingdom | |
| | | Fruits and vegetables | India | United Kingdom | |
| | *Salmonella* | Nuts, nut products and seeds | India | NDA | |
| 2016 | Aflatoxins | Fruits and vegetables Herbs and spices | Turkey India | NDA NDA | [20] |
| | | Nuts, nut products and seeds | China, Egypt, Iran, Turkey, United States | Germany, Italy, The Netherlands, United Kingdom | |
| | Pesticide residues | Fruits and vegetables | Turkey | Bulgaria, The Netherlands | |
| | *Salmonella* | Fruits and vegetables | India | United Kingdom | |
| 2017 | Aflatoxins | Fruits and vegetables Nuts, nut products and seeds | Turkey China, Iran, Turkey | NDA Germany, Italy, The Netherlands, Spain | [21] |
| | Absence of health certificate(s) | Nuts, nut products and seeds | NDA | United Kingdom | |
| | Pesticide residues | Fruits and vegetables | Turkey | NDA | |
| 2018 | Aflatoxins | Nuts, nut products and seeds | Argentina, China, Egypt, Turkey, United States | Germany, Italy, The Netherlands, Spain, United Kingdom | [22] |
| | Ochratoxin A | Fruits and vegetables | NDA | Turkey | |
| | *Salmonella* | Nuts, nut products and seeds | Sudan | Greece | |

**Table 14.** *Cont.*

| Year | Hazard | Product Category | Origin Country * | Notifying Country * | Reference |
|---|---|---|---|---|---|
| 2019 | Aflatoxins | Fruits and vegetables | Turkey | NDA | [23] |
|  |  | Nuts, nut products and seeds | Argentina, Turkey, United States | Germany, Italy, The Netherlands, Spain |  |
|  | Ochratoxin A | Fruits and vegetables | NDA | Turkey |  |
|  | *Salmonella* | Herbs and spices | Brazil | NDA |  |
|  |  | Nuts, nut products and seeds | Sudan | Greece |  |
| 2020 | Aflatoxins | Fruits and vegetables | Turkey | NDA | [24] |
|  |  | Nuts, nut products and seeds | Argentina, Iran, Turkey, United States | Germany, The Netherlands |  |
|  | Ethylene oxide | Nuts, nut products and seeds | India | Germany, The Netherlands |  |
|  | Pesticide residues | Fruits and vegetables | Turkey | Bulgaria |  |
|  | *Salmonella* | Herbs and spices | Brazil | Germany |  |

* NDA—No Data Available.

Mycotoxins (including afltotocins) are highly carcinogenic and mutagenic, and are therefore an important issue in food production [25,26]. Cereals, spices and nuts can be infected with mycotoxins [25]. It is estimated that 25% of the world's cereal production is contaminated by these compounds [25,27]. Therefore, they cause significant losses in agriculture [28], especially in developing countries [29]. Economic losses associated with mycotoxin contamination include the costs of prevention, storage of infected wastes and quality control, and are calculated at billions of euros per year [30].

Some findings presented in annual RASFF reports overlap with results from the aforementioned cluster analysis. These were pesticide residues in fruits and vegetables from Turkey (in 2016, 2017 and 2020), *Salmonella* in fruits and vegetables from India (in 2015), nuts and seeds from Sudan (in 2018 and 2109), herbs and spices from Brazil (in 2019), the absence of health certificate(s) for nuts and seeds (in 2017), and ethylene oxide in seeds from India (in 2020).

The problem with the presence of pesticide residues in fruits and vegetables has been recognised in Turkey. Therefore, in this country a pesticide monitoring programme is required due to health and environmental concerns [31]. Turkey is a world leader in fresh produce, so it is believed that a surveillance system is needed to ensure food safety [32]. It should also be mentioned that the fruit and vegetable sector is very important for Turkey because of trade with the European Union [33]. In the context of pesticide residues, it is also worth noting the presence of ethylene oxide in sesame seeds from India. This problem continued and further diversified in 2021, leading to the largest food recall in the EU's history [9]. *Salmonella* is also frequently indicated in RASFF annual reports. Due to the environmental changes in the food chain, reducing the presence of this bacterium is more difficult than laboratory tests might suggest [34]. Therefore, it is important to understand how *Salmonella* can adopt, avoid and/or suppress plant defences in order to take appropriate strategies [35].

## 4.2. RASFF Notifications in Studies by Various Authors

Various authors often referred to RASFF notifications for plants in their studies, but provided very little information. Therefore, after reviewing the studies on notifications in this system, only those with the following variables were selected: year(s), hazard or hazard category and product or product category. They were sorted by year(s) of notification and then in alphabetical order. The product names were, in fact, given by authors with varying degrees of detail and, in addition, sometimes modified. If the authors also provided the country of origin, this is indicated in brackets after the name of the product category or product (Table 15).

**Table 15.** Hazards in products of plant origin by studies of various authors on RASFF notifications.

| Year(s) | Hazard or Hazard Category | Product or Product Category | Reference |
|---|---|---|---|
| 1979–2020 | Food additives and flavourings, pathogenic micro-organisms, pesticide residues | Fruits and vegetables | [36] |
| 1979–2020 | Mycotoxins | Herbs and spices | [36] |
| 1979–2020 | Mycotoxins, pathogenic micro-organisms | Nut products and seeds | [36] |
| 1999–2020 | *Bacillus cereus*, *Clostridium* spp., *Listeria monocytogenes*, *Salmonella* spp. | Mushrooms | [37] |
| 2000–2010 | Noroviruses | Berries, Tomatoes | [38] |
| 2000–2015 | Health certificate(s), illegal importation, tampering | Cereals and bakery products, fruits and vegetables, nuts, nut products and seeds | [39] |
| 2001–2010 | Aflatoxins | Fruits, nuts (from Argentina, Brazil, China, Egypt, Ghana, India, Iran, Turkey and United States) | [40] |
| 2001–2013 | Carbendazim | Aubergines, beans, broccoli, celery, chamomile, grapes, mint, okra, papaya | [41] |
| 2001–2015 | *Listeria monocytogenes* | Fruits and vegetables (from Germany) | [42] |
| 2002–2014 | Aflatoxins | Groundnuts, hazelnuts, pistachios, figs, herbs and spices | [43] |
| 2002–2018 | Genetically modified | Linseed, maize, papaya, rice | [44] |
| 2002–2019 | Pesticides | Gherkins (from Turkey) | [31] |
| 2002–2019 | Aflatoxins | Figs, hazelnuts, pistachios (from China, Iran, Turkey, United States) | [45] |
| 2002–2020 | Pesticide residues | Fruits, vegetables, nuts | [46] |
| 2002–2020 | Pesticide residues | Apples, pomegranates, peppers (from Turkey), rice (from India), tea (from China) | [47] |
| 2003 | Aflatoxins | Maize | [48] |
| 2003–2005 | Sudan | Chilli, paprika, turmeric-derived spicy products, palm oil | [49] |
| 2003–2006 | Aflatoxins | Peanuts, tree nuts | [50] |
| 2003–2007 | Aflatoxins | Pistachios (from Iran) | [51] |
| 2003–2007 | *Escherichia coli* | Spice and condiments | [51] |
| 2003–2007 | Genetically modified | Rice (from China, United States) | [51] |
| 2003–2007 | Noroviruses | Raspberries | [51] |
| 2003–2007 | Ochratoxin A | Cereals, figs, pepper, raisins/sultanas, vegetables | [51] |
| 2003–2007 | Pesticides | Fruits and vegetables | [51] |
| 2003–2007 | Sudan 4 | Palm oil (from African countries) | [51] |
| 2003–2009 | Sudan | Palm oil (from African countries) | [52] |
| 2004–2007 | *Bacillus cereus*, *Escherichia coli*, *Listeria monocytogenes*, *Salmonella* | Mushrooms | [53] |
| 2004–2008 | Dimethoate, insect, mould, rodent excrements, *Salmonella*, sulphite | Edible flowers (from Albania, Egypt, Sri Lanka and Thailand) | [54] |
| 2004–2009 | *Salmonella* | Rucola (from Italy) | [55] |
| 2004–2013 | Genetically modified | Papaya (from China, Thailand, Vietnam, United States) | [56] |
| 2004–2014 | Aflatoxin B1, ochratoxin A | Chilli, nutmeg, paprika, pepper | [57] |
| 2004–2014 | *Salmonella* spp. | Basil, coriander, black pepper, peppermint | [57] |
| 2004–2014 | *Bacillus* spp. | Chilli, curry | [57] |
| 2004–2014 | Aflatoxins, pesticide residues, Sudan | Herbs and spices | [58] |
| 2004–2018 | Dimethoate, insects, mould, rodent excrements, *Salmonella*, sulphite | Edible flowers | [59] |
| 2005 | Aflatoxins, Ochratoxin A | Fruits and vegetables, herbs and spices, nuts and nut products (pistachios from Iran) | [60] |
| 2005 | Aflatoxins | Pistachios | [61] |
| 2005–2006 | Microbiological contamination | Herbs and spices | [62] |
| 2005–2014 | Pathogenic micro-organisms | Almonds, coconuts, hazelnuts, pine nuts, pistachios | [63] |
| 2005–2015 | Chemical contaminants, foreign bodies, mycotoxins, pesticide residues, unauthorized additives and adulteration | Fruits and vegetables (from Turkey, India and Thailand) | [64] |

| Year(s) | Hazard or Hazard Category | Product or Product Category | Reference |
|---|---|---|---|
| 2005–2020 | *Salmonella* | Parsley | [65] |
| 2006 | Genetically modified | Rice (from China) | [40] |
| 2006–2015 | Aflatoxins | Paprika | [66] |
| 2007 | *Salmonella* | Alfalfa (from Pakistan) | [67] |
| 2008 and before | Aflatoxins | Pistachios | [68] |
| 2008–2011 | Additives, bacterial pathogens, chemical hazards, heavy metals, hygiene hazard/insufficient quality, mycotoxins, pesticide residues, physical hazard, viruses | Fruits and vegetables | [69] |
| 2008–2011 | bacterial pathogens, hygiene hazard/insufficient quality, mycotoxins, pesticide residues | Herbs and spices | [69] |
| 2008–2011 | Bacterial pathogens, hygiene hazard/insufficient quality, mycotoxins, genetically modified | Nuts, nut products and seeds | [69] |
| 2009 | Norovirus | Raspberries | [70] |
| 2009 and before | Aflatoxins, ochratoxin A | Cereals | [71] |
| 2009–2012 | Norovirus | Raspberries, strawberries | [72] |
| 2010–2011 | Aflatoxins | Nuts, nut products and seeds | [73] |
| 2010–2011 | Genetically modified | Rice (from China) | [74] |
| 2010–2012 | Norovirus, hepatovirus A | Dates (from Algeria), lettuce (from France, Germany), raspberries (from Chile, China, Poland, Serbia) | [75] |
| 2010–2014 | Norovirus, hepatovirus A | Dates (from Algeria), lettuce (from France), raspberries (from Chile, China, Poland, Serbia) | [76] |
| 2011 | Aflatoxins | Groundnuts | [77] |
| 2011 | *Salmonella*, *Escherichia coli* | Betel (from Bangladesh, India and Thailand) | [40] |
| 2011 | Aflatoxins, Ochratoxin A | herbs and spices, fruits and vegetables, nuts, nut products and seeds | [67] |
| 2011 | Norovirus | Raspberries | [78] |
| 2011 and before | Microbiological hazards | Fruits and vegetables, herbs and spices | [70] |
| 2011–2012 | Aflatoxins, ochratoxin A | Cereals and bakery products | [30] |
| 2011–2013 | Norovirus, hepatovirus A | Raspberries, strawberries | [79] |
| 2011–2014 | Allergens | Cereals and bakery products, cocoa, cocoa preparations, coffee and tea, fruits and vegetables, herbs and spices, nuts, nut products and seeds | [80] |
| 2011–2017 | Allergens | Cereals and bakery products | [81] |
| 2012 | Aflatoxins | Hazelnuts, figs, pistachios | [33] |
| 2012 | Pesticide residues | Pepper | [33] |
| 2012 | Genetically modified | Rice (from China) | [82] |
| 2012 | Allergens | Wheat | [83] |
| 2012 and before | Pesticide residues | Tea | [84] |
| 2012–2015 | Aflatoxins | Maize (from Bulgaria, Croatia, Greece, Hungary, Italy, Serbia, Slovakia, Poland, Romania) | [26] |
| 2012–2017 | *Salmonella* | Herbs and spices, nuts, nut products and seeds | [85] |
| 2012–2021 | Pyrrolizidine alkaloids | Spices and aromatic herbs, tea | [86] |
| 2013 | Mycotoxins, pathogenic micro-organisms, pesticide residues | Fruits and vegetables | [87] |
| 2013–2014 | Carbendazim | Mint | [88] |
| 2014 | Norovirus | Raspberries, strawberries | [89] |
| 2014 | Aflatoxins | Nuts and nut products | [90] |
| 2014–2018 | Chlorpyrifos | Herbs and spices | [91] |
| 2015 and before | Aflatoxins | Chilli | [92] |
| 2015–2018 | Norovirus, Hepatovirus A | Strawberries | [93] |

**Table 15.** *Cont.*

| Year(s) | Hazard or Hazard Category | Product or Product Category | Reference |
|---|---|---|---|
| 2015–2020 | Pesticide residues | Fruits and vegetables | [94] |
| 2016 | Aflatoxins | Nuts | [95] |
| 2016 | Mycotoxins | Herbs | [96] |
| 2016 | Mycotoxins | Herbs and spices, nuts, nut products and seeds | [97] |
| 2017 | Aflatoxins | Fruits and vegetables, herbs and spices, Nuts, nut products and seeds (from India) | [98] |
| 2017 | Aflatoxins | Nuts, nut products and seeds | [99] |
| 2017 and before | Pesticide residues | Chilli, paprika | [100] |
| 2017 and before | Additives | Chilli, curcuma, curry, palm oil, pepper | [101] |
| 2017 and before | Sudan | Herbs and spices | [102] |
| 2017–2021 | Aflatoxins, ochratoxin A, insects, missing documents, pesticides, sulphites | Figs (from Turkey and Spain) | [103] |
| 2018 | Enteric viruses | Berries | [104] |
| 2019 | Aflatoxins | Nuts | [105] |
| 2019 | Aflatoxins | Nuts | [106] |
| 2019 | Ochratoxin A | Figs, raisins | [106] |
| 2019 | Chlorpyrifos | Fruits and vegetables | [106] |
| 2019 and before | Mycotoxins | Maize, rice, wheat | [107] |
| 2020–2022 | Aflatoxins, ochratoxin A | Figs (from Turkey) | [108] |

The studies carried out by these authors confirm the results presented in Section 3 (Results) regarding the three most frequently reported hazard categories in the RASFF. It was noted that the notifications mainly concerned mycotoxins (aflatoxins and ochratoxin A) in fruits and vegetables, herbs and spices, and nuts. Another hazard category was pesticide residues (including, e.g., carbendazim, chlorpyrifos, dimethoate) notified in fruits and vegetables and herbs and spices. A third clearly noticeable hazard category was pathogenic micro-organisms (including *Escherichia coli*, *Listeria monocytogenes* and *Salmonella* spp.), which were similarly reported in fruits and vegetables and herbs and spices and, to a lesser extent, also in nuts.

Attention was also paid to RASFF notifications of other hazards (most of which were presented in the Section 3): additives including Sudan dye in herbs and spices and palm oil, genetically modified rice, foreign bodies in fruits and vegetables, lack of health certificates for fruits and vegetables, herbs and spices, and nuts or allergens in cereals and bakery products. However, the authors also highlighted hazards reported in the RASFF in other products: pesticides (dimethoate), foreign bodies, *Salmonella* spp. and sulphites in edible flowers, pathogenic micro-organisms (including *Bacillus cereus*, *Clostridium* spp., *Escherichia coli*, *Listeria monocytogenes*, *Salmonella* spp.) in mushrooms, and norovirus and hepatovirus A in strawberries and raspberries.

When the origin of the notified products was indicated, they were mainly Asian countries (Turkey, India, China, Thailand), the United States, and African and South American countries.

## 5. Conclusions

The three most commonly encountered hazards in foods of plant origin, i.e., mycotoxins, pesticide residues and pathogenic micro-organisms (including microbial contamination) related to 80% of notifications in the European Rapid Alert System for Food and Feed (RASFF) in 1998–2020. Particular attention should be paid to the hazards that have occurred in recent years: pesticide residues in peppers, moulds in groundnuts, ochratoxin A in raisins and sulphite in apricots from Turkey, ethylene oxide in sesame and problems with health certificate(s) for chilli, nutmeg, pistachios and sesame from India, iodine in seaweed from China and South Korea, *Salmonella* in sesame from Brazil, India and Sudan, *Escherichia coli* in basil from Lao Republic and betel from Thailand, and colour in breakfast cereals from the United States.

The notified products were, therefore, mainly from non-EU countries (particularly from Asia), i.e., Turkey, followed by India, China and Iran, and also from the United States. Given their proximity to the EU common market, hazards in products from Turkey (which shares a land border with Bulgaria and Greece) are of particular concern. These products were reported on the basis of border rejections, information notifications and, to a lesser extent, alerts. Notifications were based on border control, after which the consignments were detained, or official controls placed on the market; consequently, products were re-dispatched, withdrawn or destroyed.

Measures leading to the elimination of unsafe food products of plant origin from the European Union common market were necessary, but resulted in high costs and image losses for farmers, producers and other economic operators. Therefore, farmers need to pay particular attention to the use of methods such as Good Agricultural Practice (GAP), Good Hygiene Practice (GHP) and Good Manufacturing Practice (GMP), because, through these methods, hazards in food products of plant origin can be largely prevented or eliminated. It is also important that pesticides used by farmers to reduce or suppress the presence of pathogenic micro-organisms and the effects of their activities should be applied in an appropriate and proportionate manner, and with withdrawal periods. Producers (processors) should be more involved in the control of fresh produce delivered by farmers. Transporters should pay attention to maintaining the right parameters (temperature and humidity), especially in sea transport from distant Asian countries. It is also important for hazard limits to be set and updated by legislative bodies, and subsequently controlled by the authorities of the EU countries.

The "From field to fork" strategy adopted in the European Green Deal emphasises the need to build a sustainable model in the food system, and the elimination or reduction of hazards in plants is an important part of this strategy. Therefore, the research carried out, covering a wide time period and range of hazards found in food products of plant origin, can contribute to improvements in sustainability efforts.

**Supplementary Materials:** The following supporting information can be downloaded at: https://www.mdpi.com/article/10.3390/su15108091/s1, Table S1: Number of notifications in the RASFF under product categories for the period 1979–2020; Table S2: Hazard categories and hazards notified in products of plant origin in the RASFF in 1998–2020; Table S3: Shortened values names of variables notification basis, distribution status and action taken used in figures in Supplementary Materials; Figure S1: Results of joining cluster analysis; (a) year; (b) product; (c) notifying country; (d) origin country; (e) notification type; (f) notification basis; (g) distribution status; (h) action taken; Figure S2: Results of two-way joining cluster analysis for aflatoxins; (a) product; (b) notifying country; (c) origin country; (d) notification type; (e) notification basis; (f) distribution status; (g) action taken; Figure S3: Results of two-way joining cluster analysis for ochratoxin A; (a) product; (b) notifying country; (c) origin country; (d) notification type; (e) notification basis; (f) distribution status; (g) action taken; Figure S4: Results of two-way joining cluster analysis for ethylene oxide; (a) product; (b) notifying country; (c) origin country; (d) notification type; (e) notification basis; (f) distribution status; (g) action taken; Figure S5: Results of two-way joining cluster analysis for chlorpyrifos; (a) product; (b) notifying country; (c) origin country; (d) notification type; (e) notification basis; (f) distribution status; (g) action taken; Figure S6: Results of two-way joining cluster analysis for carbendazim; (a) product; (b) notifying country; (c) origin country; (d) notification type; (e) notification basis; (f) distribution status; (g) action taken; Figure S7: Results of two-way joining cluster analysis for dimethoate; (a) product; (b) notifying country; (c) origin country; (d) notification type; (e) notification basis; (f) distribution status; (g) action taken; Figure S8: Results of two-way joining cluster analysis for methomyl; (a) product; (b) notifying country; (c) origin country; (d) notification type; (e) notification basis; (f) distribution status; (g) action taken; Figure S9: Results of two-way joining cluster analysis for acetamiprid; (a) product; (b) notifying country; (c) origin country; (d) notification type; (e) notification basis; (f) distribution status; (g) action taken; Figure S10: Results of two-way joining cluster analysis for omethoate; (a) product; (b) notifying country; (c) origin country; (d) notification type; (e) notification basis; (f) distribution status; (g) action taken; Figure S11: Results of two-way joining cluster analysis for triazophos; (a) product; (b) notifying country; (c) origin country; (d) notification type; (e) notification basis; (f) distribution status; (g) action taken; Figure S12: Results of two-way

joining cluster analysis for formetanate; (a) product; (b) notifying country; (c) origin country; (d) notification type; (e) notification basis; (f) distribution status; (g) action taken; Figure S13: Results of two-way joining cluster analysis for *Salmonella*; (a) product; (b) notifying country; (c) origin country; (d) notification type; (e) notification basis; (f) distribution status; (g) action taken; Figure S14: Results of two-way joining cluster analysis for *Escherichia coli*; (a) product; (b) notifying country; (c) origin country; (d) notification type; (e) notification basis; (f) distribution status; (g) action taken; Figure S15: Results of two-way joining cluster analysis for moulds; (a) product; (b) notifying country; (c) origin country; (d) notification type; (e) notification basis; (f) distribution status; (g) action taken; Figure S16: Results of two-way joining cluster analysis for Sudan; (a) product; (b) notifying country; (c) origin country; (d) notification type; (e) notification basis; (f) distribution status; (g) action taken; Figure S17: Results of two-way joining cluster analysis for iodine; (a) product; (b) notifying country; (c) origin country; (d) notification type; (e) notification basis; (f) distribution status; (g) action taken; Figure S18: Results of two-way joining cluster analysis for sulphite; (a) product; (b) notifying country; (c) origin country; (d) notification type; (e) notification basis; (f) distribution status; (g) action taken; Figure S19: Results of two-way joining cluster analysis for colour; (a) product; (b) notifying country; (c) origin country; (d) notification type; (e) notification basis; (f) distribution status; (g) action taken; Figure S20: Results of two-way joining cluster analysis for genetically modified plants; (a) product; (b) notifying country; (c) origin country; (d) notification type; (e) notification basis; (f) distribution status; (g) action taken; Figure S21: Results of two-way joining cluster analysis for insects; (a) product; (b) notifying country; (c) origin country; (d) notification type; (e) notification basis; (f) distribution status; (g) action taken; Figure S22: Results of two-way joining cluster analysis for health certificate(s); (a) product; (b) notifying country; (c) origin country; (d) notification type; (e) notification basis; (f) distribution status; (g) action taken; Figure S23: Results of two-way joining cluster analysis for milk; (a) product; (b) notifying country; (c) origin country; (d) notification type; (e) notification basis; (f) distribution status; (g) action taken.

**Author Contributions:** Conceptualization, methodology, formal analysis, investigation, data curation, writing—original draft preparation, M.P.; writing—review and editing, M.N.-D. All authors have read and agreed to the published version of the manuscript.

**Funding:** The APC was funded by Gdynia Maritime University, Department of Quality Management, team research project "Systemic quality, environment and safety management in the product life cycle" number WZNJ/2023/PZ/04 and Cracow University of Economics, Department of Quality Management.

**Institutional Review Board Statement:** Not applicable.

**Informed Consent Statement:** Not applicable.

**Data Availability Statement:** Not applicable.

**Conflicts of Interest:** The authors declare no conflict of interest.

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
