# Peer review of "Hazards in Products of Plant Origin Reported in the Rapid Alert System for Food and Feed (RASFF) from 1998 to 2020"

_sustainability, doi:10.3390/su15108091_

Round 1
Reviewer 1 Report
The review of the RASFF results is interesting and meaningful. It shows the wide spectrum of the results and the situations of food hazards. It shows the results of hazards of plant origin what makes it valuable as usualy more information can be found about the animal origin. The orginality of the paper is average as this subject of RASFF report is quite popular.
Author Response
Dear Reviewer,
Thank You very much for positive comments on the submitted paper.
Reviewer 2 Report
Relevance of the study:
This investigation is highly relevant, focusing on the hazards occurring in foods of plant origin. However, this study would be expected to fall into the scope of journals like food control, food policy, or even the MDPI journal Foods. It is a fact that can be related with sustainability, but I would not expect this paper in this journal. Nevertheless, if the editor believes the paper can fit the scope of this journal, on a broad sense, then I will not oppose.
Abstract:
If the work is to possibly be published in a journal focusing on sustainability, I believe that the abstract should better emphasize in which way this research on food control can be framed into sustainability issues, definitely. One other aspect is that the abstract does not provide any insights at all about the methodology followed to undertake the research, how did the authors obtain the data they used for their treatment – the source? This is mandatory to be mentioned briefly in the abstract. Also, in concluding, they should emphasize the contribution of this work to the global sustainability.
Introduction:
The introduction helps to frame and contextualize the work but lacks a supporting Literature Review. The justification of study is provided, including the graphs that help understand why the time frame was chose after 1998, but there is no literature review with studies about related work, namely the hazards and their effects, on all perspectives, like health or economic. This should be brief, but be present in the introduction, even though the authors used this in their discussion part in more detail. I believe this part needs some improvements as well.
Materials and methods:
The description of the obtaining of the data and their subsequent analysis is comprehensive and I do not have suggestions for improvement in this part.
Results and discussion:
The results are presented clearly and due to the high volume of results presented in Tables and Figures, the authors opted to provide most of them as supplementary material, which was a correct decision.
Conclusions:
The conclusions part is also well formulated, presenting the most relevant findings of the work, and, unlike in the abstract this part concludes with a note about the usefulness on these restlst as a way to support sustainable food chains.
Use of English is fair
Author Response
Dear Reviewer,
Thank You very much for all your comments on the submitted paper. I have included these comments below in CAPITAL LETTERS and my responses in plain font. Changes in the paper have been made using the “Track Changes” function.
COMMENTS AND SUGGESTIONS FOR AUTHORS
RELEVANCE OF THE STUDY:
THIS INVESTIGATION IS HIGHLY RELEVANT, FOCUSING ON THE HAZARDS OCCURRING IN FOODS OF PLANT ORIGIN. HOWEVER, THIS STUDY WOULD BE EXPECTED TO FALL INTO THE SCOPE OF JOURNALS LIKE FOOD CONTROL, FOOD POLICY, OR EVEN THE MDPI JOURNAL FOODS. IT IS A FACT THAT CAN BE RELATED WITH SUSTAINABILITY, BUT I WOULD NOT EXPECT THIS PAPER IN THIS JOURNAL. NEVERTHELESS, IF THE EDITOR BELIEVES THE PAPER CAN FIT THE SCOPE OF THIS JOURNAL, ON A BROAD SENSE, THEN I WILL NOT OPPOSE.
In order to better justify that the safety of products of plant origin falls within the scope of sustainability, the last sentence of the Conclusions has been moved to the Abstract. In the Conclusions, however, this sentence has been expanded.
ABSTRACT:
IF THE WORK IS TO POSSIBLY BE PUBLISHED IN A JOURNAL FOCUSING ON SUSTAINABILITY, I BELIEVE THAT THE ABSTRACT SHOULD BETTER EMPHASIZE IN WHICH WAY THIS RESEARCH ON FOOD CONTROL CAN BE FRAMED INTO SUSTAINABILITY ISSUES, DEFINITELY. ONE OTHER ASPECT IS THAT THE ABSTRACT DOES NOT PROVIDE ANY INSIGHTS AT ALL ABOUT THE METHODOLOGY FOLLOWED TO UNDERTAKE THE RESEARCH, HOW DID THE AUTHORS OBTAIN THE DATA THEY USED FOR THEIR TREATMENT – THE SOURCE? THIS IS MANDATORY TO BE MENTIONED BRIEFLY IN THE ABSTRACT. ALSO, IN CONCLUDING, THEY SHOULD EMPHASIZE THE CONTRIBUTION OF THIS WORK TO THE GLOBAL SUSTAINABILITY.
Abstract has been expanded and improved.
INTRODUCTION:
THE INTRODUCTION HELPS TO FRAME AND CONTEXTUALIZE THE WORK BUT LACKS A SUPPORTING LITERATURE REVIEW. THE JUSTIFICATION OF STUDY IS PROVIDED, INCLUDING THE GRAPHS THAT HELP UNDERSTAND WHY THE TIME FRAME WAS CHOSE AFTER 1998, BUT THERE IS NO LITERATURE REVIEW WITH STUDIES ABOUT RELATED WORK, NAMELY THE HAZARDS AND THEIR EFFECTS, ON ALL PERSPECTIVES, LIKE HEALTH OR ECONOMIC. THIS SHOULD BE BRIEF, BUT BE PRESENT IN THE INTRODUCTION, EVEN THOUGH THE AUTHORS USED THIS IN THEIR DISCUSSION PART IN MORE DETAIL. I BELIEVE THIS PART NEEDS SOME IMPROVEMENTS AS WELL.
Between points 1 and 1.1, issues related to sustainability, food security and food safety have been added, taking into account several new sources where they have been addressed.
MATERIALS AND METHODS:
THE DESCRIPTION OF THE OBTAINING OF THE DATA AND THEIR SUBSEQUENT ANALYSIS IS COMPREHENSIVE AND I DO NOT HAVE SUGGESTIONS FOR IMPROVEMENT IN THIS PART.
As recommended by Reviewer 3, subsection "2.1.1. Hazards analysed" has been rewritten and corrected.
RESULTS AND DISCUSSION:
THE RESULTS ARE PRESENTED CLEARLY AND DUE TO THE HIGH VOLUME OF RESULTS PRESENTED IN TABLES AND FIGURES, THE AUTHORS OPTED TO PROVIDE MOST OF THEM AS SUPPLEMENTARY MATERIAL, WHICH WAS A CORRECT DECISION.
Following the recommendation of Reviewer 4, the results of the cluster analysis contained in the tables were moved from the Supplementary Material to the Results section, but all figures remained there.
CONCLUSIONS:
THE CONCLUSIONS PART IS ALSO WELL FORMULATED, PRESENTING THE MOST RELEVANT FINDINGS OF THE WORK, AND, UNLIKE IN THE ABSTRACT THIS PART CONCLUDES WITH A NOTE ABOUT THE USEFULNESS ON THESE RESTLST AS A WAY TO SUPPORT SUSTAINABLE FOOD CHAINS.
As recommended by Reviewer 3, the Conclusions have been slightly reworded and shortened, and the individual paragraphs more closely linked to each other.
COMMENTS ON THE QUALITY OF ENGLISH LANGUAGE
USE OF ENGLISH IS FAIR
Reviewer 3 Report
In general, the topic of the manuscript is very interesting and the information shared is important. However, there were some problems that need the authors' attention.
1. The authors should revise the text for grammatical accuracy and review word choices. Several sentences are difficult to understand and need to be rephrased/reworded. Wordiness is common and makes the text boring. I have highlighted the most problematic parts (in my opinion), see attached file.
2. Clarify the difference between alert notifications and information notifications, especially information notifications (the sentence that explains it is contradictory).
3. The discussion section is weak and needs to be improved. The way it is right now, it's just restating/rewriting the results.
4. The conclusion needs to be improved, it does not flow well, and the transitions between paragraphs are not there, so paragraphs are disconnected.

Sentence structure, grammar, and word choice must be improved.
Author Response
Dear Reviewer,
Thank You very much for all your comments on the submitted paper. I have included these comments below in CAPITAL LETTERS and my responses in plain font. Changes in the paper have been made using the “Track Changes” function.
COMMENTS AND SUGGESTIONS FOR AUTHORS
IN GENERAL, THE TOPIC OF THE MANUSCRIPT IS VERY INTERESTING AND THE INFORMATION SHARED IS IMPORTANT. HOWEVER, THERE WERE SOME PROBLEMS THAT NEED THE AUTHORS' ATTENTION.
- THE AUTHORS SHOULD REVISE THE TEXT FOR GRAMMATICAL ACCURACY AND REVIEW WORD CHOICES. SEVERAL SENTENCES ARE DIFFICULT TO UNDERSTAND AND NEED TO BE REPHRASED/REWORDED. WORDINESS IS COMMON AND MAKES THE TEXT BORING. I HAVE HIGHLIGHTED THE MOST PROBLEMATIC PARTS (IN MY OPINION), SEE ATTACHED FILE.
Sentences and phrases have been reworded or corrected.
- CLARIFY THE DIFFERENCE BETWEEN ALERT NOTIFICATIONS AND INFORMATION NOTIFICATIONS, ESPECIALLY INFORMATION NOTIFICATIONS (THE SENTENCE THAT EXPLAINS IT IS CONTRADICTORY).
This fragment related to information notification was corrected: “Information notifications are used when a risk in food or feed has been identified but other RASFF members do not need to take rapid action because the product has not reached their market or is no longer on their market or the nature of the risk does not require rapid action”.
- THE DISCUSSION SECTION IS WEAK AND NEEDS TO BE IMPROVED. THE WAY IT IS RIGHT NOW, IT'S JUST RESTATING/REWRITING THE RESULTS.
Two paragraphs have been added below Table 14 relating to the most important hazards, i.e. mycotoxins, pesticide residues and Salmonella.
- THE CONCLUSION NEEDS TO BE IMPROVED, IT DOES NOT FLOW WELL, AND THE TRANSITIONS BETWEEN PARAGRAPHS ARE NOT THERE, SO PARAGRAPHS ARE DISCONNECTED.
The conclusions have been shortened and corrected. Individual paragraphs have been reworded and linked together.
COMMENTS ON THE QUALITY OF ENGLISH LANGUAGE
SENTENCE STRUCTURE, GRAMMAR, AND WORD CHOICE MUST BE IMPROVED.
The article has been corrected in accordance with the Reviewer's recommendations regarding the English language and then reviewed again.
Reviewer 4 Report
Line 11, Abstract: delete notifications, as it is repeated
Keywords: I don’t think “ European Union” is appropriate as it is too general; I suggest replacing it with “chemical hazard” for example
Line 16: “The following are very important” please rephrase as it is not at all scientific, e.g. specify if “the following” are your conclusions or something else, “important” for who/ what, etc.
Line 24: instead of “was mentioned” “is mentioned”
Line 26: without “there”
Lines 27-30: I suggest combining the two phrases: “The European Union law attaches considerable importance to food safety. Already more than 40 years ago, the Rapid Alert System for Food and Feed (RASFF) was established to provide information on food safety risks, which occurred in the food chain.”
Line 30-31: “The largest number of notifications in this system concerns plants.” This information is not explained enough, it is not clear if this is your conclusion or one you have read: please indicate which is its source, to which period it refers, etc.
If this is a conclusion of your work, please clarify the following aspect: your results indicate in lines 61-63 that 43% of the total notification refer to plant origin products, which is not really a majority.
Line 34-35: “Regulation (EC) No 178/2002 of the European Parliament and of the Council of 28 34 January 2002” EC refers to European Council, there is no need for the explanation, so “Regulation (EC) No 178/2002” is enough
Line 37: I suggest replacing “Its members” with “RASFF system is based on / involves …”
Line 39-41: this idea should be mentioned above, it belongs in lines before 36/37
Line 42-43: “Alert notifications are sent when food or feed presenting a serious risk is already on the market and rapid action is required.”
This is not entirely true for two reasons:
1. RASSF notifications can be issued after a regular check at the EU border, so not only when the product is on the market, situation which is also presented in lines 49-51;
2. RASSF alerts can be issued for a potential hazard, not only when the hazard is clearly identified/ confirmed.
Lines 46-49: “Information notifications are used if a risk has been identified in a product on the market, but no rapid action is required by other RASFF members. This happens when the product has not reached their markets, is no longer pre sent on their markets or the nature of the risk does not require rapid action” This is illogical: first sentence refers to a product on the market and the second one, an explanatory one of the first sentence, refers to “the product has not reached their markets, is no longer present on their markets” Please explain or rephrase
Line 58-59: “all product categories that appear in the RASFF were shown in Table S1 in the Supplementary Material” As the name of the table indicates it does not refer to the whole database, but for the period 1979-2020, so this should be specified in the text e.g. “RASSF in the analysed period”
Lines 66-68: “By far the largest number of notifications concerned nuts, nut products and seeds and fruits and vegetables. In 2009, a decrease in the number of notifications for both categories can be observed.”
The results are not scientifically presented, they should be interpreted in comparison to a reference or with numerical/ statistical data interpretation (e.g. “a 32% decrease”)
Line 16: “i.e. 68% of the initial population” Please explain if population refers to a statistical one or something else
Line 134-135: “the names have been changed to those used in the British English (and so: cabbages to squashes”
Please clarify this information because cabbage (latin name Brassica oleracea) is different from squashes (Cucurbita genus)!!!!
Line 307: “Notifications on mycotoxins related products from Asian countries.” I suggest add “to products”
Line 308: “as many as half” Replace this with the scientific way to express the exact number of %
Lines 313: “A more contemporary problem” referring to 2016-2019 compared to the period analyzed before, 2003-2006. I suggest changing the expression as many of the readers are contemporary to both periods, not only one (e.g. The four years between 2016-2019 …”)
Line 319: “Pesticide residues were the most numerous group of reported hazards (nine different 319 substances in the analysed population).” Indicate exact numbers to support your conclusion!
Line 322-323: “by Bulgaria in recent years” I suggest change it to “several years”, as 2010-2011 is not really recent
“Another country that frequently appeared in the notifications was India.” Add “pesticide related notifications”
Line 329: “In turn” is not suited here, delete it
Line 351: Escherichia coli should be italic
Figures and tables of statistical analysis of the data presented in the article should not be in the Supplementary file, but in the main body of the manuscript
Author Response
Dear Reviewer,
Thank You very much for all your comments on the submitted paper. I have included these comments below in CAPITAL LETTERS and my responses in plain font. Changes in the paper have been made using the “Track Changes” function.
COMMENTS AND SUGGESTIONS FOR AUTHORS
LINE 11, ABSTRACT: DELETE NOTIFICATIONS, AS IT IS REPEATED
Deleted
KEYWORDS: I DON’T THINK “ EUROPEAN UNION” IS APPROPRIATE AS IT IS TOO GENERAL; I SUGGEST REPLACING IT WITH “CHEMICAL HAZARD” FOR EXAMPLE
Replaced by the following phrase: “food hazard”
LINE 16: “THE FOLLOWING ARE VERY IMPORTANT” PLEASE REPHRASE AS IT IS NOT AT ALL SCIENTIFIC, E.G. SPECIFY IF “THE FOLLOWING” ARE YOUR CONCLUSIONS OR SOMETHING ELSE, “IMPORTANT” FOR WHO/ WHAT, ETC.
Replaced by the phrase “In order to ensure the safety of food of plant origin it is necessary to”
LINE 24: INSTEAD OF “WAS MENTIONED” “IS MENTIONED”
Changed
LINE 26: WITHOUT “THERE”
Deleted
LINES 27-30: I SUGGEST COMBINING THE TWO PHRASES: “THE EUROPEAN UNION LAW ATTACHES CONSIDERABLE IMPORTANCE TO FOOD SAFETY. ALREADY MORE THAN 40 YEARS AGO, THE RAPID ALERT SYSTEM FOR FOOD AND FEED (RASFF) WAS ESTABLISHED TO PROVIDE INFORMATION ON FOOD SAFETY RISKS, WHICH OCCURRED IN THE FOOD CHAIN.”
These two sentences have been reworded and combined: “The European Union attaches particular importance to food safety, therefore already in 1979 the Rapid Alert System for Food and Feed (RASFF) was established to provide information on risks in the food chain.”
LINE 30-31: “THE LARGEST NUMBER OF NOTIFICATIONS IN THIS SYSTEM CONCERNS PLANTS.” THIS INFORMATION IS NOT EXPLAINED ENOUGH, IT IS NOT CLEAR IF THIS IS YOUR CONCLUSION OR ONE YOU HAVE READ: PLEASE INDICATE WHICH IS ITS SOURCE, TO WHICH PERIOD IT REFERS, ETC.
The sentence has been rewritten and expanded, and a source has been added: “During the period 1979-2020, the largest number of notifications in this system related to food of plant origin (more than 43%), followed by food of animal origin (30%), with the remaining notifications referring to other types of food, feed and food contact materials [2].”
IF THIS IS A CONCLUSION OF YOUR WORK, PLEASE CLARIFY THE FOLLOWING ASPECT: YOUR RESULTS INDICATE IN LINES 61-63 THAT 43% OF THE TOTAL NOTIFICATION REFER TO PLANT ORIGIN PRODUCTS, WHICH IS NOT REALLY A MAJORITY.
I did not use the word “majority” but the phrase “largest number”. However, this sentence was indeed not clear enough, so I completed it with additional information and I believe that there is now no contradiction between these two sentences.
LINE 34-35: “REGULATION (EC) NO 178/2002 OF THE EUROPEAN PARLIAMENT AND OF THE COUNCIL OF 28 34 JANUARY 2002” EC REFERS TO EUROPEAN COUNCIL, THERE IS NO NEED FOR THE EXPLANATION, SO “REGULATION (EC) NO 178/2002” IS ENOUGH
The phrase “of the European Parliament and of the Council of 28 January 2002” was deleted.
LINE 37: I SUGGEST REPLACING “ITS MEMBERS” WITH “RASFF SYSTEM IS BASED ON / INVOLVES …”
Replaced by the phrase “The members of the system are:”
LINE 39-41: THIS IDEA SHOULD BE MENTIONED ABOVE, IT BELONGS IN LINES BEFORE 36/37
The position of the sentence has been changed.
LINE 42-43: “ALERT NOTIFICATIONS ARE SENT WHEN FOOD OR FEED PRESENTING A SERIOUS RISK IS ALREADY ON THE MARKET AND RAPID ACTION IS REQUIRED.”
THIS IS NOT ENTIRELY TRUE FOR TWO REASONS:
- RASSF NOTIFICATIONS CAN BE ISSUED AFTER A REGULAR CHECK AT THE EU BORDER, SO NOT ONLY WHEN THE PRODUCT IS ON THE MARKET, SITUATION WHICH IS ALSO PRESENTED IN LINES 49-51;
- RASSF ALERTS CAN BE ISSUED FOR A POTENTIAL HAZARD, NOT ONLY WHEN THE HAZARD IS CLEARLY IDENTIFIED/ CONFIRMED.
On the European Commission’s website, with regard to alert notifications, only the information concerning the product present on the market is given. However, it is indeed indirectly implicit in Article 50 of Regulation 178/2002 that these two other possibilities also exist, therefore this information has been added.
LINES 46-49: “INFORMATION NOTIFICATIONS ARE USED IF A RISK HAS BEEN IDENTIFIED IN A PRODUCT ON THE MARKET, BUT NO RAPID ACTION IS REQUIRED BY OTHER RASFF MEMBERS. THIS HAPPENS WHEN THE PRODUCT HAS NOT REACHED THEIR MARKETS, IS NO LONGER PRE SENT ON THEIR MARKETS OR THE NATURE OF THE RISK DOES NOT REQUIRE RAPID ACTION” THIS IS ILLOGICAL: FIRST SENTENCE REFERS TO A PRODUCT ON THE MARKET AND THE SECOND ONE, AN EXPLANATORY ONE OF THE FIRST SENTENCE, REFERS TO “THE PRODUCT HAS NOT REACHED THEIR MARKETS, IS NO LONGER PRESENT ON THEIR MARKETS” PLEASE EXPLAIN OR REPHRASE
This is exactly how it was put down on the European Commission’s website. However, the information does indeed appear to contradict each other, therefore these two sentences have been reworded and combined: “Information notifications are used when a risk in food or feed has been identified but other RASFF members do not need to take rapid action because the product has not reached their market or is no longer on their market or the nature of the risk does not require rapid action.”.
LINE 58-59: “ALL PRODUCT CATEGORIES THAT APPEAR IN THE RASFF WERE SHOWN IN TABLE S1 IN THE SUPPLEMENTARY MATERIAL” AS THE NAME OF THE TABLE INDICATES IT DOES NOT REFER TO THE WHOLE DATABASE, BUT FOR THE PERIOD 1979-2020, SO THIS SHOULD BE SPECIFIED IN THE TEXT E.G. “RASSF IN THE ANALYSED PERIOD”
The phrase “in the period 1979-2020” was added.
LINES 66-68: “BY FAR THE LARGEST NUMBER OF NOTIFICATIONS CONCERNED NUTS, NUT PRODUCTS AND SEEDS AND FRUITS AND VEGETABLES. IN 2009, A DECREASE IN THE NUMBER OF NOTIFICATIONS FOR BOTH CATEGORIES CAN BE OBSERVED.”
THE RESULTS ARE NOT SCIENTIFICALLY PRESENTED, THEY SHOULD BE INTERPRETED IN COMPARISON TO A REFERENCE OR WITH NUMERICAL/ STATISTICAL DATA INTERPRETATION (E.G. “A 32% DECREASE”)
These two sentences have been completed and reworded: “By far the largest number of notifications concerned nuts, nut products and seeds and fruits and vegetables (37% and 35%, respectively, of all notifications to plants between 1979 and 2020). Between 2009 and 2010, a 27% decrease in the number of notifications to nuts, nut products and seeds can be observed, and in 2009 a 12% decrease in the number of notifications to fruit and vegetables can be seen.”
LINE 16: “I.E. 68% OF THE INITIAL POPULATION” PLEASE EXPLAIN IF POPULATION REFERS TO A STATISTICAL ONE OR SOMETHING ELSE
This sentence was reworded: “These were 22 hazards, for which 22,687 notifications were made between 1998 and 2020 (68 % of the notifications on plants during this period)”.
LINE 134-135: “THE NAMES HAVE BEEN CHANGED TO THOSE USED IN THE BRITISH ENGLISH (AND SO: CABBAGES TO SQUASHES”
PLEASE CLARIFY THIS INFORMATION BECAUSE CABBAGE (LATIN NAME BRASSICA OLERACEA) IS DIFFERENT FROM SQUASHES (CUCURBITA GENUS)!!!!
The RASFF database rarely provides the Latin names of products. However, indeed cabbages and squashes are two different product species. Therefore, they have been separated and the figures in the Supplementary Material have also been corrected accordingly.
LINE 307: “NOTIFICATIONS ON MYCOTOXINS RELATED PRODUCTS FROM ASIAN COUNTRIES.” I SUGGEST ADD “TO PRODUCTS”
Corrected
LINE 308: “AS MANY AS HALF” REPLACE THIS WITH THE SCIENTIFIC WAY TO EXPRESS THE EXACT NUMBER OF %
This part of the text was reworded:
“Notifications relating to mycotoxins (aflatoxins and ochratoxin A) are presented in Table 5.”
“These notifications were reported most frequently and accounted for up to 55% of the notifications examined using two-way joining cluster analysis. These mainly concerned products from Asia, but aflatoxins in pistachios from Iran were the most prominent problem.”
LINES 313: “A MORE CONTEMPORARY PROBLEM” REFERRING TO 2016-2019 COMPARED TO THE PERIOD ANALYZED BEFORE, 2003-2006. I SUGGEST CHANGING THE EXPRESSION AS MANY OF THE READERS ARE CONTEMPORARY TO BOTH PERIODS, NOT ONLY ONE (E.G. THE FOUR YEARS BETWEEN 2016-2019 …”)
The sentence was reworded: “Ochratoxin A in raisins from Turkey was notified in 2016-2019.”.
LINE 319: “PESTICIDE RESIDUES WERE THE MOST NUMEROUS GROUP OF REPORTED HAZARDS (NINE DIFFERENT 319 SUBSTANCES IN THE ANALYSED POPULATION).” INDICATE EXACT NUMBERS TO SUPPORT YOUR CONCLUSION!
The phrase was reworded: “Notifications related to pesticide residues are presented in Table 6. It was the most numerous group of reported hazards (9 different substances out of the 22 hazards analysed).”.
LINE 322-323: “BY BULGARIA IN RECENT YEARS” I SUGGEST CHANGE IT TO “SEVERAL YEARS”, AS 2010-2011 IS NOT REALLY RECENT
Corrected
“ANOTHER COUNTRY THAT FREQUENTLY APPEARED IN THE NOTIFICATIONS WAS INDIA.” ADD “PESTICIDE RELATED NOTIFICATIONS”
The sentence was reworded: “Another country that frequently appeared in notifications relating to pesticide residues was India.”.
LINE 329: “IN TURN” IS NOT SUITED HERE, DELETE IT
Deleted
LINE 351: ESCHERICHIA COLI SHOULD BE ITALIC
Corrected
FIGURES AND TABLES OF STATISTICAL ANALYSIS OF THE DATA PRESENTED IN THE ARTICLE SHOULD NOT BE IN THE SUPPLEMENTARY FILE, BUT IN THE MAIN BODY OF THE MANUSCRIPT
Table S4 has been moved from the Supplementary Material to the manuscript (now Table 4). Table S5 has been moved to the manuscript and divided into Tables 5-13 (depending on the hazard category). However, the figures containing the individual panels remained in the Supplementary Material as they take up dozens of pages. Table 4 and Tables 5-13 provide information referring to the relevant panels of figures in the Supplementary Material.
Round 2
Reviewer 2 Report
The authors submitted a revised version of their manuscript that was substancially improved and adresses all my recommendations. Therefore I agree to the aceptance of the mansucrpt in the current form.
Reviewer 3 Report
Thank you for addressing the issues with the first version of the manuscript. It reads much better now. The information provided is very important and should be published.
Reviewer 4 Report
All suggestions have been addressed and the manuscript is emproved, so I agree with its publication.